# Lagrangian transport simulations using the extreme convection parametrization: an assessment for the ECMWF reanalyses

Lars Hoffmann[1,3], Paul Konopka[2,3], Jan Clemens[1,2,3], and Bärbel Vogel[2,3]

[1]Jülich Supercomputing Centre, Forschungszentrum Jülich, Jülich, Germany
[2]Institut für Energie- und Klimaforschung (IEK-7), Forschungszentrum Jülich, Jülich, Germany
[3]Center for Advanced Simulation and Analytics (CASA), Forschungszentrum Jülich, Jülich, Germany

**Correspondence:** Lars Hoffmann (l.hoffmann@fz-juelich.de)

**Abstract.** Atmospheric convection plays a key role in tracer transport from the planetary boundary layer to the free troposphere. Lagrangian transport simulations driven by meteorological fields from global models or reanalysis products, such as the European Centre for Medium-Range Weather Forecasts' (ECMWF's) ERA5 and ERA-Interim reanalysis, typically lack proper explicit representations of convective up- and downdrafts because of the limited spatiotemporal resolution of the meteorology. Lagrangian transport simulations for the troposphere can be improved by applying parametrizations to better represent the effects of unresolved convective transport in the global meteorological reanalyses. Here, we implemented and assessed the effects of the extreme convection parametrization (ECP) in the Massive Parallel Trajectory Calculations (MPTRAC) model. The ECP is conceptually simple. It requires the convective available potential energy (CAPE) and the height of the equilibrium level (EL) as input parameters. Assuming that unresolved convective events yield well-mixed vertical columns of air, the ECP randomly redistributes the air parcels vertically between the surface and the EL, if CAPE is present. We analyzed statistics of explicitly resolved and parametrized convective updrafts and found that the frequencies of strong updrafts due to the ECP, i. e., 20 K potential temperature increase over 6 h or more, increase by 2 to 3 orders of magnitude for ERA5 and 3 to 5 orders of magnitude for ERA-Interim compared to the explicitly resolved updrafts. To assess the effects of the ECP on tropospheric tracer transport, we conducted transport simulations for the artificial tracer e90, which is released globally near the surface and which has a constant e-folding lifetime of 90 days throughout the atmosphere. The e90 simulations were conducted for the year 2017 with both, ERA5 and ERA-Interim. Next to sensitivity tests on the choice of the CAPE threshold, an important tuning parameter of the ECP, we suggest a modification of the ECP method, i. e., to take into account the convective inhibition (CIN) indicating the presence of warm, stable layers that prevent convective updrafts in the real atmosphere. While ERA5 has higher spatiotemporal resolution and explicitly resolves more convective updrafts than ERA-Interim, we found there is still a need for both reanalyses to apply a convection parametrization such as the ECP to better represent tracer transport from the planetary boundary layer into the free troposphere on the global scale.

## 1 Introduction

Convection plays a key role in the dynamics and composition of the Earth's atmosphere via affecting its heat, moisture and momentum budgets (Emanuel, 1994; Stevens, 2005). Convection is triggered by air parcel – environment instability, i. e., due

to different lapse rates within dry and moist air masses. If the environmental lapse rate is steeper than the lapse rate experienced by a rising air parcel, upward-displaced air parcels become buoyant and experience further upward force. Intense moist convection can lead to thunderstorm development and severe weather conditions, such as flash flooding, gust fronts, or tornadoes. Deep convection is a particularly relevant and rapid process for transport of air from the planetary boundary layer into the free troposphere and lower stratosphere (Dickerson et al., 1987; Fischer et al., 2003; Monks et al., 2009). Collectively, convective features are important drivers of the Hadley and Walker circulations, the monsoon circulations, and the El Niño–Southern Oscillation. Accurate representation of convection is a prerequisite for reliable numerical weather prediction and climate modelling. A standard approach to represent convection in numerical models is to apply convection parametrizations (Tiedtke, 1989; Kain and Fritsch, 1993; Bechtold et al., 2004, 2008, 2014).

Lagrangian transport models are indispensable tools to simulate and analyze atmospheric transport processes in research and operational applications (Draxler and Hess, 1998; McKenna et al., 2002a,b; Lin et al., 2003; Stohl et al., 2005; Jones et al., 2007; Lin et al., 2012; Stein et al., 2015; Sprenger and Wernli, 2015; Pisso et al., 2019). The models are often driven by wind fields and other meteorological variables from global reanalyses or forecasts. More recently, the European Centre for Medium-Range Weather Forecasts (ECMWF) released its state-of-the-art ERA5 reanalysis (Hersbach et al., 2020). Based on ten years of continuous developments, ERA5 was significantly improved compared to its predecessor ERA-Interim (Dee et al., 2011), including better physical parametrizations in the forecast model and improvements in the data assimilation scheme and the observations. Hoffmann et al. (2019) showed that the transition from ERA-Interim to ERA5 yields significant improvements in Lagrangian transport simulations. For instance, ERA5 trajectory calculations for the stratosphere showed much better conservation of potential temperature along the trajectories than corresponding ERA-Interim calculations. Furthermore, Hoffmann et al. (2019) found that the representation of convective updrafts in the troposphere was improved from ERA-Interim to ERA5, which was largely attributed to the increased spatiotemporal resolution of ERA5. Similarly, Li et al. (2020) showed that convective transport in tropical cyclones was improved from ERA-Interim to ERA5.

Using global meteorological fields from reanalyses and forecasts to drive Lagrangian transport simulations is a suitable solution in many applications. However, the spatial and temporal resolution of the global input fields is often still too coarse to allow for explicit representation of convective up- and downdrafts. Convective updrafts are typically confined to horizontal scales well below a few kilometers. In order to achieve mass balancing, the updrafts are compensated by both, intense small-scale downdrafts and large-scale subsidence of air. Next to the limited spatial resolution, convective up- and downdrafts occur on a timescale of a few hours or less, which is not well-represented in 6-hourly output time intervals of reanalyses such as ERA-Interim. Considering that convection is a subgrid-scale process, individual convective events are typically not well represented in coarse resolution global meteorological models. Therefore, various techniques and parametrizations have been developed to better represent the effects of convection in Lagrangian transport simulations for the troposphere.

Brinkop and Jöckel (2019) introduce a new version of the Atmospheric Tracer Transport in a LAgrangian model (ATTILA) scheme, which includes a newly developed parameterisation for Lagrangian convection. The Lagrangian convection scheme of ATTILA uses the mass fluxes of a standard convection scheme (Tiedtke, 1989; Nordeng, 1994) implemented in a Eulerian general circulation model, which drives the simulations, to calculate the movement of convective parcels. Convection is initiated

when the convergence of moisture in a vertical column of the atmosphere exceeds a certain threshold and a convectively unstable layer is present. In the Lagrangian convection scheme of ATTILA, air parcels can follow the updraft, downdraft, or compensating motion in the environment at a grid column with convection within a time step. The forcing used for the Lagrangian convection scheme is provided by the mass fluxes of the convection scheme, which are converted into probabilities to trigger up- or downward motions of the air parcels involved in the parametrized convection.

Wohltmann et al. (2019) present the Lagrangian convective transport scheme LaConTra developed for global chemistry and transport models. Similar to ATTILA, the LaConTra scheme is driven by convective mass fluxes and detrainment rates that originate from external convection parameterizations of general circulation models that are used to produce meteorological re-analyses or forecasts. LaConTra builds upon the current approach of Lagrangian convective transport schemes to stochastically redistribute air parcels within a fixed time step according to estimated probabilities for convective entrainment as well as the altitude of detrainment. The new approach of LaConTra allows for modeling the variable time that an air parcel spends in convection by estimating vertical updraft velocities, which are obtained by combining convective mass fluxes from meteorological fields with a parameterization of convective area fraction profiles. The scheme considers the variable residence time that an air parcel spends in convection, which is particularly important for accurately simulating the tropospheric chemistry of short-lived species.

Konopka et al. (2019) discuss tropospheric mixing and the parametrization of unresolved convective updrafts as implemented in version 2.0 of the Chemical Lagrangian Model of the Stratosphere (CLaMS). Considering some deficiencies in the representation of the effects of convective uplifting and mixing due to weak vertical stability in the troposphere, the CLaMS transport scheme was modified by including additional tropospheric mixing and vertical transport due to unresolved convective updrafts by parametrizing these processes in terms of the dry and moist Brunt–Väisälä frequencies. The scheme enhances upward transport for conditionally unstable air parcels from the lowest model layer into the upper troposphere. Using the diabatic concept of vertical motion, the uplift is represented by estimating and adding a change in potential temperature $\Delta\theta$ to the air parcel trajectories such that stable conditions for the air parcel are fulfilled.

Here, we assess the extreme convection parametrization (ECP) introduced by Gerbig et al. (2003) as implemented and applied in the Stochastic Time-Inverted Lagrangian Transport model (STILT) and Hybrid Single-Particle Lagrangian Integrated Trajectory model (HYSPLIT). The ECP method assumes that convective up- and downdrafts are generally large enough to leave a perfectly well-mixed column behind each convective event. The method vertically mixes all air parcels in grid cells with positive convective available potential energy (CAPE) throughout the unstable layer between the surface and the equilibrium level. Despite being less sophisticated than more advanced convection parametrization schemes applied in other models, the ECP is conceptually simple and very easy to implement in a Lagrangian model. It is also computationally cheap and poses no significant overhead on the total runtime of the Lagrangian transport simulations, which is an advantage for conducting large-scale ensemble or long-term simulations.

In this study, we assess the effects of applying the ECP on Lagrangian transport simulations to properly represent global tracer transport in the free troposphere and stratosphere. Noting that convective transport in the troposphere is generally under-estimated in coarse-resolution, global reanalysis horizontal wind and vertical velocity fields driving the Lagrangian transport

simulations, the ECP is expected to mitigate these limitations. We conduct our assessment of the ECP using two ECMWF reanalyses, the state-of-the-art ERA5 reanalysis and its predecessor ERA-Interim, in order to evaluate how the ECP simulations are affected by the different driving meteorological fields. In particular, by systematically comparing ECP and non-ECP simulations with ERA5 and ERA-Interim, we aim to show that while there are large differences in explicitly resolved convective transport between ERA5 and ERA-Interim, both of which significantly underestimate the amount of convective transport in the real atmosphere, the ECP largely mitigates these problems by contributing significantly larger numbers of parameterised convective updrafts to the transport simulations, to a level comparable between ERA5 and ERA-Interim.

First, we compared statistical distributions of explicitly resolved and parametrized convective updrafts for both, ERA5 and ERA-Interim. This comparison is based on an analysis of potential temperature change of air parcel trajectories lifted upwards on convective timescales from the model lower boundary layer into the free troposphere. Second, we assessed the impact of applying the ECP on global transport simulations of the artificial tracer e90, which is released globally near the surface and has constant e-folding lifetime of 90 days throughout the atmosphere. By definition, e90 is particularly well suited to assess tropospheric transport processes (Prather et al., 2011; Abalos et al., 2017). We conducted various sensitivity tests of ECP and non-ECP simulations driven by ERA5 and ERA-Interim in order to test the robustness of the results and to provide guidance for parameter choices for the ECP. In this paper, we also propose a possible enhancement of the ECP method by considering the convective inhibition (CIN) to hinder physically unrealistic parametrized convection events in the presence of stable, blocking layers.

In Sect. 2, we provide brief descriptions of the ECMWF reanalyses, the MPTRAC Lagrangian transport model, the ECP as implemented in MPTRAC, and the model settings applied for the statistical analysis and sensitivity tests. Section 3 provides the results of the study, including illustrative examples of particle dispersion simulations with or without applying the ECP, a statistical analysis of convective updrafts as well as the e90 artificial tracer transport simulations. Finally, Sect. 4 provides a short summary and the conclusions of the study.

## 2 Methods

### 2.1 The ECMWF reanalyses

In this study, we conducted the trajectory calculations and Lagrangian particle dispersion simulations with meteorological fields from ECMWF reanalyses. The fifth-generation reanalysis ERA5 (Hersbach et al., 2020) is produced with ECMWF's Integrated Forecasting System (IFS) Cycle 41r2 as implemented in March 2016. ERA5 provides hourly estimates of various atmospheric, terrestrial, and oceanic climate variables. Atmospheric variables are available at a horizontal resolution of $\sim$31 km ($T_L639$ spectral grid). Here, we retrieved the fields on an $0.3° \times 0.3°$ longitude-latitude grid from ECMWF's Meteorological Archival and Retrieval System (MARS). The ERA5 data are provided on 137 hybrid sigma-pressure levels, with the top level being located at 0.01 hPa or about 80 km of altitude. ERA5 covers the time period from January 1950 to present.

For some cases, we also conducted the Lagrangian transport simulations with meteorological fields from the ERA-Interim reanalysis (Dee et al., 2011), being the predecessor of ERA5. ERA-Interim was produced using IFS Cycle 31r2 of December

2006. It is based on 4-dimensional variational analysis (4D-Var) with a 12-hour analysis window. The ERA-Interim data are provided 6-hourly for the time period from January 1979 to August 2019. The horizontal resolution of ERA-Interim is ∼79 km

($T_L 255$ spectral grid). It covers 60 model levels from the surface up to 0.1 hPa or about 65 km of altitude. We retrieved the fields on an $0.75° \times 0.75°$ longitude-latitude grid on all model levels from MARS. From the technical specifications, it can be seen that ERA5 improves upon ERA-Interim in various aspects. The spatiotemporal resolution is much better, but also the physical parametrizations of various processes were significantly improved in the newer version of the IFS model used for ERA5 (Hennermann and Berrisford, 2018; Hersbach et al., 2020).

As part of the data processing for the Lagrangian transport simulations with MPTRAC, we vertically interpolated the ECMWF meteorological fields from model levels to pressure levels using the Climate Data Operators (CDO, Schulzweida, 2014). For the vertical interpolation, the number of the target pressure levels and their spacing have been chosen to correspond to the original IFS model levels, using ECMWF's $a$ and $b$ coefficients for the L137 (ERA5) and L60 (ERA-Interim) model level definitions, respectively. In addition, a constant surface pressure of 1013.25 hPa was used to determine the target pressure levels

for interpolation. Transferring the ECMWF analyses from model levels to pressure levels introduces small interpolation errors. However, we could not avoid this vertical interpolation as the MPTRAC model in its present form requires the meteorological fields to be given on pressure levels.

## 2.2  The MPTRAC Lagrangian transport model

Massive-Parallel Trajectory Calculations (MPTRAC) is a Lagrangian particle dispersion model for the analysis of atmospheric

transport processes in the free troposphere and stratosphere (Hoffmann et al., 2016, 2022a). The model found applications in several case studies addressing long-range transport simulations and the estimation of sulfur dioxide injections from volcanic eruptions (Heng et al., 2016; Wu et al., 2017, 2018; Cai et al., 2022). More recently, Zhang et al. (2020) and Smoydzin and Hoor (2022) used MPTRAC to study aerosol and trace gas variations in the upper troposphere and lower stratosphere over Asia and the remote Pacific. The model was also used for the evaluation of meteorological reanalyses and trajectory calculations using

superpressure balloon observations (Hoffmann et al., 2017). It is important to note that MPTRAC is not targeting boundary layer applications. At present, the model lacks proper representation of more complex mixing and diffusion processes in the boundary layer and it applies pressure as vertical coordinate, which is not terrain-following and therefore less suited for the boundary layer.

MPTRAC calculates air parcel trajectories using 4-D linear interpolation of given wind fields and the explicit midpoint

method for numerical integration of the kinematic equations of motion (Rößler et al., 2018). Mesoscale diffusion and subgrid-scale wind fluctuations are simulated using a Langevin equation to add stochastic perturbations to the trajectories, closely following the approach applied in the FLEXPART model (Stohl et al., 2005; Pisso et al., 2019). Additional modules are implemented in MPTRAC to simulate convection, sedimentation, radioactive decay, hydroxyl chemistry, dry deposition, and wet deposition. MPTRAC provides various output methods for the air parcel data, including grid-averaged output, which is

applied in this study. While MPTRAC can be applied on a single workstation, it features an MPI-OpenMP-OpenACC hybrid

parallelization for efficient use on high performance computing (HPC) systems and graphics processing units (GPUs) (Rößler et al., 2018; Liu et al., 2020; Hoffmann et al., 2022a).

Next to the convection module, which is described in more detail in Sect. 2.3, we applied another module of MPTRAC, which allows for sampling of various meteorological variables along the trajectories. We used this module to determine potential temperature along the trajectories. From the potential temperature change over a given time period, which was selected as 6 h to match the order of typical convective timescales (Keil et al., 2014; Bullock et al., 2015; Konopka et al., 2022), we calculated statistics of explicitly resolved and parametrized convective updrafts in ERA5 and ERA-Interim (Sect. 3.2). The e90 artificial tracer simulations were conducted using the boundary condition module of MPTRAC to prescribe the e90 concentrations of the air parcels at the lower boundary of the model and a module to simulate exponential loss of the concentrations using a given, fixed e-folding lifetime (Sect. 3.3 and following).

## 2.3 The extreme convection parametrization

Atmospheric convection is characterized by various fundamental physical quantities. Among the most important quantities are the convective available potential energy and the convective inhibition (Blanchard, 1998; Riemann-Campe et al., 2009). The convective available potential energy (CAPE) is the integrated amount of work that upward buoyancy force can perform on an air parcel if it rose vertically through the atmosphere. CAPE is calculated from

$$\text{CAPE} = -\int\limits_{p_{\text{LFC}}}^{p_{\text{EL}}} R_d(T_{vp} - T_{ve})d\ln p. \tag{1}$$

where $p$ is pressure, $T_{vp}$ is the virtual temperature of the lifted air parcel moving upward moist adiabatically from the level of free convection (LFC) to the equilibrium level (EL), $T_{ve}$ is the virtual temperature of the environment, and $R_d$ is the specific gas constant for dry air. Similarly, the convective inhibition (CIN) is calculated from

$$\text{CIN} = \int\limits_{p_{\text{SFC}}}^{p_{\text{LFC}}} R_d(T_{vp} - T_{ve})d\ln p. \tag{2}$$

Conceptually, CIN is the opposite of CAPE. It indicates the amount of energy that will prevent an air parcel from rising from the surface (SFC) via the lifted condensation level to the LFC. Physically, CIN indicates the presence of warm, stable layers that will effectively hinder the formation of convective updrafts. Both, the definitions of CAPE and CIN, take into account the virtual temperature, $T_v = T(1 + \epsilon q)$, with temperature $T$, $\epsilon = 0.608$, and specific humidity $q$. The virtual temperature is the temperature that dry air would have if its pressure and density would match a given sample of moist air. In the definitions of CAPE and CIN, the use of virtual temperature reduces uncertainties due to neglecting the effects of moisture on the equation of state (Doswell and Rasmussen, 1994).

As an example, Fig. 1 shows monthly mean CAPE, CIN, LFC, and EL fields in July 2017 from the ERA5 reanalysis. The variables have been calculated directly from the ERA5 temperature and specific humidity vertical profiles using the meteorological data preprocessing code of MPTRAC (Hoffmann et al., 2022a, see electronic supplement). CAPE is largest in the

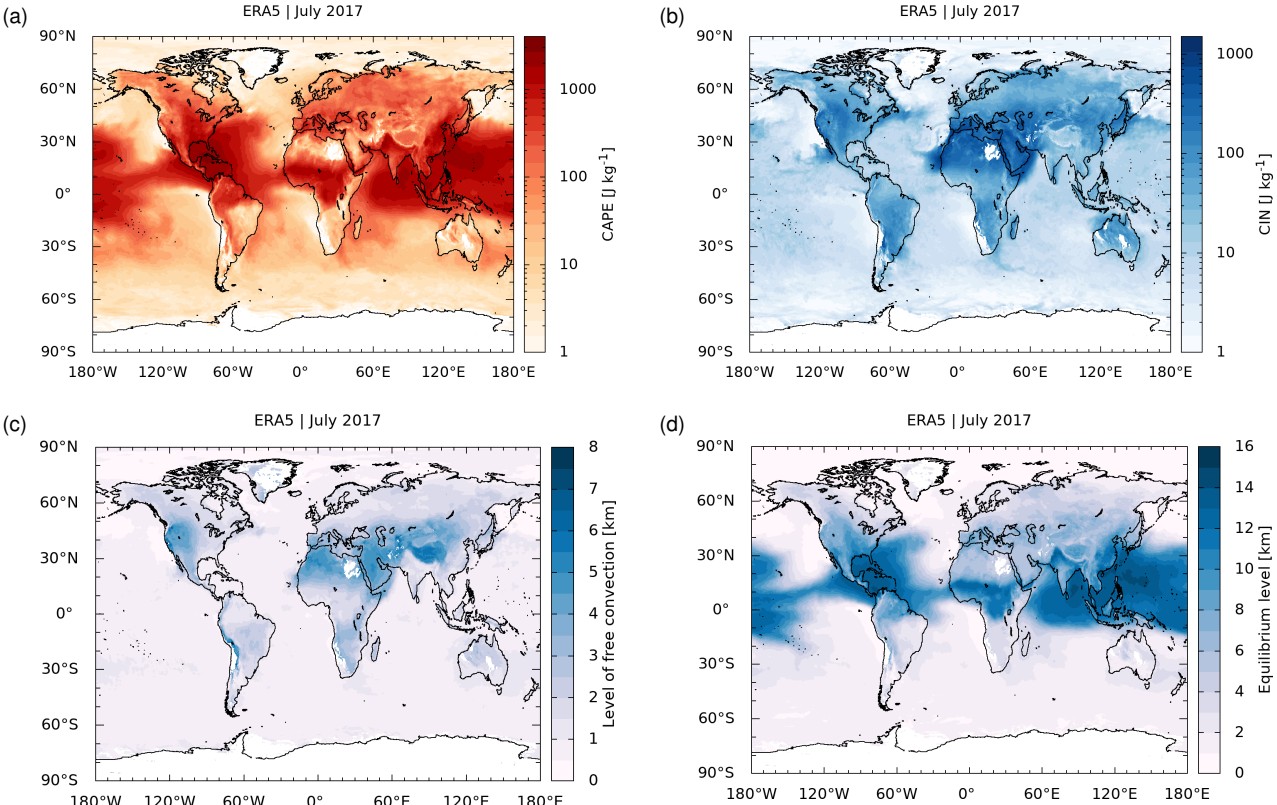

**Figure 1.** Monthly mean convective available potential energy (CAPE, a), convective inhibition (CIN, b), log-pressure height of the level of free convection (LFC, c), and log-pressure height of the equilibrium level (EL, d) from ERA5 for July 2017. White areas over hot or cold deserts indicate regions where CAPE and CIN are absent.

tropics, near the Intertropical Convergence Zone (ITCZ), where high temperature and moisture strongly promote convection. Local means of CAPE of up to 3000 J kg$^{-1}$ relate to frequent events of intense convection. CAPE decreases by 2 to 3 orders of magnitude at higher latitudes. CAPE minima are observed over cold water ocean surfaces and in arid regions over land. In contrast, the largest values of CIN are found over the subtropics, near the downwelling regions of the Hadley circulation. CIN is stronger over land than over ocean. A broad maximum of CIN, with peak values of up to 1400 J kg$^{-1}$ in the monthly mean occurs over Northern Africa and the Arabian Peninsula. The patterns found here are qualitatively consistent with other studies discussing climatologies and long-term changes of CAPE and CIN (Riemann-Campe et al., 2009; Chen et al., 2020).

The ECP as implemented in MPTRAC requires CAPE and EL derived from the meteorological fields. In the first step, the gridded CAPE and EL values are interpolated to the horizontal positions of the air parcels. If the interpolated CAPE value of an air parcel is larger than a threshold CAPE$_0$, i.e., a user-defined control parameter of the ECP, and the air parcel is located below the EL, it is assumed that up- and downdrafts within the vertical column are strong enough to trigger a convective event. In the second step, the air parcels involved in a convective event are randomly redistributed in the vertical column stretching

from the surface to the EL. The random redistribution of the air parcels is weighted by air density in order to yield a well-mixed vertical column of air over the grid boxes of the meteorological fields. Mass conservation is achieved because the number of air parcels and their mass are not changed in a convective mixing event. In the following time step of the model, the trajectories are continued from the new vertical positions the air parcels were assigned to during the convective mixing event. In MPTRAC, the ECP can be applied as frequent as each time step of the model, or it can be applied more sparsely at user-defined time intervals (e. g., every ~3 h) to reflect typical convective timescales.

The globally applied threshold $CAPE_0$ can be set to zero, implying that convection will take place everywhere below the EL where CAPE exists. Strictly speaking, this parameter choice is referred to as the "extreme convection" approach. It provides an upper limit to the effects of unresolved convection in the meteorological fields. In contrast, switching off the ECP completely will provide a lower limit for the effects of convection on the Lagrangian transport simulations, as only explicitly resolved convective updrafts of the meteorological fields will be taken into account. Intermediate states can be simulated by selecting specific values of the threshold $CAPE_0$. As there is no fixed classification, we here refer to CAPE values of less than $\sim 1000\,\mathrm{J\,kg^{-1}}$ to represent weak to moderate instability, $\sim 1000$ to $\sim 3000\,\mathrm{J\,kg^{-1}}$ to represent moderate to strong instability, and CAPE values greater than $\sim 3000\,\mathrm{J\,kg^{-1}}$ to indicate cases of extreme instability.

To provide guidance on choosing the threshold $CAPE_0$ for the ECP, Fig. 2a shows occurrence frequencies of convective events and the frequency distributions of CAPE values exceeding a given threshold in different latitude bands derived from global ERA5 data on 1 July 2017, 00:00 UTC. Similar to Fig. 1, it is found that convective events are predominant in the tropics, followed by middle latitudes, whereas strong CAPE events are much less frequent at high latitudes. Despite the large variability, Fig. 2b shows that the mean height of the EL tends to scale logarithmically with CAPE, increasing from mean heights of about $2\,\mathrm{km}$ below $10\,\mathrm{J\,kg^{-1}}$ to about $14\,\mathrm{km}$ for CAPE values larger than $1000\,\mathrm{J\,kg^{-1}}$. This correlation seems noteworthy, as it might potentially be used estimate the height of the EL from CAPE fields, in case this information is missing in the meteorological fields. For example, for the ECMWF reanalyses the EL fields from the CAPE calculations are not available from the MARS archive, which is why we applied the MPTRAC meteorological data pre-processing code to obtain this information. However, as various additional processes and parameters affect the individual distributions of CAPE and EL, the idea needs to be further investigated in future work. In Sect. 3.5, we will discuss sensitivity tests showing how different choices of $CAPE_0$ impact tracer transport simulations.

Here, we presented examples of CAPE and CIN fields derived from the ERA5 reanalysis, but we also conducted comparisons with the corresponding ERA-Interim fields. These comparisons generally revealed good agreement of the convective variables between the reanalyses, which is promising, as similar CAPE and EL fields from ERA5 and ERA-Interim are a prerequisite to yield similar results in ECP transport simulations. If the CAPE and EL fields derived from ERA5 and ERA-Interim are similar, the parametrized convective updrafts of the ECP are not expected to largely differ between the reanalyses. The differences between using ERA5 and ERA-Interim to drive ECP transport simulations are further discussed in Sect. 3.4.

While CAPE is a key parameter for the ECP, the onset and characteristics of convective activity are also characterized by various other variables. Here, we propose a modification of the ECP method by considering the convective inhibition (CIN) in addition to CAPE when triggering convective events. CIN can be used to detect cases where layers of warm air yield stability,

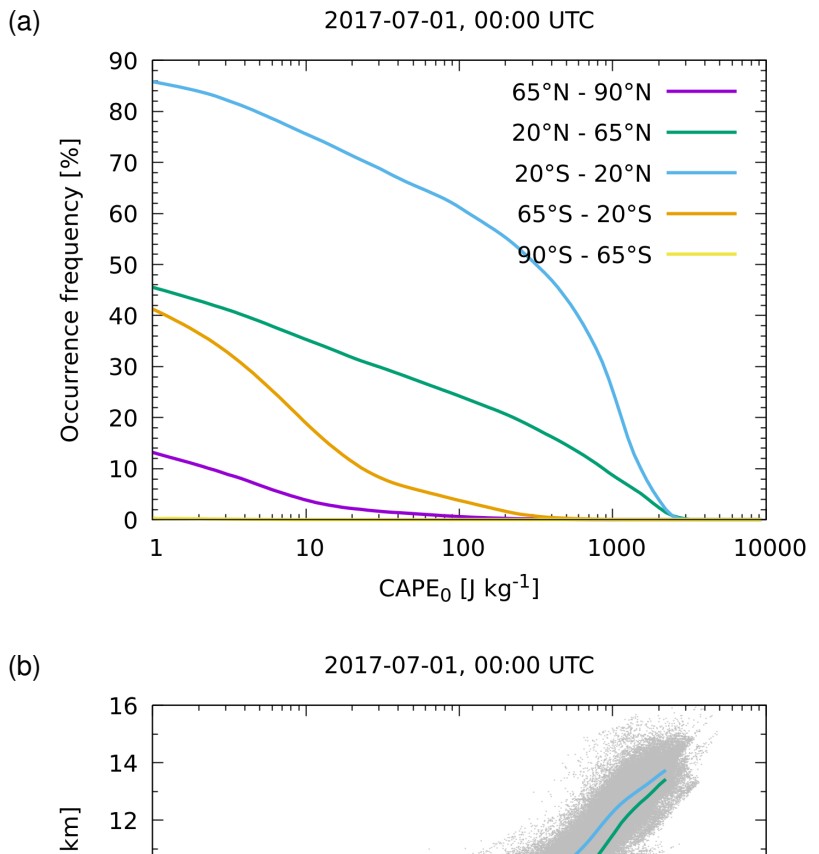

**Figure 2.** Cumulative distribution functions of (a) convective events exceeding a given threshold $CAPE_0$ as shown along the x-axis and (b) mean EL versus CAPE from ERA5 on 1 July 2017, 00:00 UTC in different latitude bands (see plot key). Gray dots in panel (b) show CAPE and EL values of individual $0.3° \times 0.3°$ horizontal grid boxes of the reanalysis fields.

preventing cooler air parcels from rising in the atmosphere. CIN indicates the amount of energy needed to force air parcels to push through and rise above a stable layer. CIN is typically stronger over land than over ocean and shows the largest means
and variability over the subtropics, in particular over Northern Africa and the Arabian Peninsula in boreal summer (Fig. 1). Considering CIN in the ECP potentially has large local effects on the transport simulations in these regions. To take into account the CIN in ECP simulations with MPTRAC, we implemented a control parameter $CIN_0$, which suppresses convective events if $CIN > CIN_0$. A sensitivity test on the choice of $CIN_0$ is discussed in Sect. 3.6.

## 2.4 Model settings for statistical analyses and sensitivity tests

In this section, we describe the different model settings and test configurations that were applied to obtain the illustrative examples of the ECP method, the statistical analysis of the explicitly resolved and parametrized convective updrafts as well as the e90 artificial tracer simulations with the ECP. First, in Sect. 3.1, we illustrate the effects of the ECP on Lagrangian particle dispersion simulations with MPTRAC using ERA5 and ERA-Interim meteorological fields in an example. In these simulations, $10^6$ particles were launched over a longitude-latitude box comprising Africa and the Atlantic Ocean (10°S to 30°N, 30°W to
20°E). This region is characterized by frequent deep convective events and local maxima in CAPE (compare Fig. 1a). The trajectory seeds had random vertical positions in the pressure range $[p_s - 150\,\text{hPa}, p_s]$ with respect to the surface pressure $p_s$ to achieve quasi-homogeneous coverage of the layer. The layer depth of 150 hPa was chosen to roughly match typical depths of the planetary boundary layer. It corresponds to a vertical extent of about 1.1 km at the standard surface pressure of 1013.25 hPa. All particles were launched on 1 July 2017, 00:00 UTC. The ECP was applied with a threshold of $CAPE_0 = 1000\,\text{J}\,\text{kg}^{-1}$ at
3-hourly time intervals. These parameter settings might be considered representative for a typical application of the ECP.

Second, in Sect. 3.2, we analyzed the statistical distributions of explicitly resolved and parametrized convective updrafts of ERA5 and ERA-Interim. The model settings and test configuration for this analysis follow the study of Konopka et al. (2022). The statistics of the updrafts are based on the analysis of trajectory calculations obtained with the MPTRAC model. Trajectories were launched every 6 h over a time period of 30 days, starting on 1 July 2017, 00:00 UTC. The trajectories cover
a time period of 6 h, i.e., the potential temperature change $\Delta\theta$ per 6 h time interval can be calculated directly from the initial and final positions of the trajectories. Positive values of $\Delta\theta$ per 6 h identify updrafts. The trajectory seeds were distributed on an $0.3° \times 0.3°$ longitude-latitude grid matching and fully covering the horizontal resolution of ERA5. In the vertical, the seeds were distributed randomly over a 150 hPa layer with respect to the surface pressure. All calculations use the same set of trajectory seeds. The ECP cases discussed here have been calculated with a threshold of $CAPE_0 = 0$ and an event frequency
matching the time step of the model ($\Delta t = 180\,\text{s}$) unless explicitly stated otherwise.

Third, in Sects. 3.3 to 3.7, we discuss transport simulations of the artificial tracer e90 to quantify the impact of the ECP on the tracer transport. In particular, we compare ECP and non-ECP simulations of the artificial tracer e90 driven by ERA5 and ERA-Interim in Sects. 3.3 and 3.4, respectively. We also discuss two sensitivity tests regarding the ECP control parameters $CAPE_0$ in Sect. 3.5 and $CIN_0$ in Sect. 3.6. The ECP simulations presented here were mostly conducted with thresholds of $CAPE_0 = 0$
and $CIN_0 = 0$, representing the case of extreme convection with parametrized convective events taking place whenever CAPE

exists, unless noted differently, when conducting the specific sensitivity tests. The model time step ($\Delta t = 180\,\text{s}$) was considered as event frequency.

The artificial tracer e90 is a passive tracer in the upper troposphere and lower stratosphere, which is of particular interest for studies related to the chemical tropopause and stratosphere-troposphere exchange (Prather et al., 2011; Abalos et al., 2017)
as well as model validation (Eyring et al., 2013; Orbe et al., 2018). The tracer e90 is emitted uniformly at the surface with a volume mixing ratio of 150 ppbv and has a constant e-folding lifetime of 90 days throughout the atmosphere. With this lifetime, e90 becomes well-mixed quickly in the troposphere. However, the lifetime is much shorter than typical timescales of stratospheric transport. The tracer e90 exhibits sharp gradients across the tropopause. The 90 ppbv contour surface of e90 is considered as a proxy of the chemical tropopause (Prather et al., 2011). By definition, the artificial tracer e90 has similar
characteristics to carbon monoxide, being a 'real' chemical tracer of atmospheric transport in the troposphere.

We initialized the e90 tracer simulations with MPTRAC by globally distributing air parcels in the pressure range from the surface up to 20 hPa (about 60 km of altitude). In the horizontal, the density of the air parcel was weighted with cosine of latitude, to achieve a quasi-homogeneous distribution of the air parcels and the mass. In the vertical, a uniform random distribution over height was applied. With this approach, near-homogeneous global coverage of the air parcels is achieved.
Each air parcel is assigned a volume mixing ratio of the tracer e90, representing the concentration of e90 in an infinitesimally small neighborhood. The e90 concentration in a larger region, e. g., for a zonal mean, is calculated by averaging the volume mixing ratios of the air parcels located in that region. The mean volume mixing ratio might be undefined, if no air parcels are located in a given volume. However, in our analysis we found that the air parcels were usually well distributed and no data gaps occurred. A total number of $10^6$ air parcels was considered for the simulations.

The initial e90 volume mixing ratio of all air parcels was set to zero. During the course of the simulation, the boundary condition module of MPTRAC was used to set the e90 volume mixing ratio in a near-surface layer to 150 ppbv. The boundary condition for e90 was prescribed at each time step of the model. Note that while for an Eulerian model the term "near-surface layer" in the definition of the e90 artificial tracer might be taken as the lowermost vertical level of the model, for a Lagrangian model the depth of the layer needs to be specified. Here, we selected the lowermost 150 hPa with respect to the surface pressure
to define the near-surface layer, thereby also following the orography. A sensitivity test on the depth of the near-surface layer is presented in Sect. 3.7. Only above the layer, the volume mixing ratios of the air parcels decay exponentially according to the prescribed 90 day e-folding lifetime of the e90 tracer. Considering this lifetime, a spin-up time of several months is required before e90 is properly distributed from the surface throughout the free troposphere. Our simulations for the year 2017 have been initialized on 1 July 2016, 00:00 UTC, aiming for a spin-up time of six months.

As earlier Lagrangian transport studies found strong effects on the choice of depth of the model lower boundary layer on simulated transport in the troposphere and lower stratosphere region (Konopka et al., 2022), we tested the sensitivity of our simulation results on the depth of the near-surface layer where the e90 volume mixing ratios of the air parcels are being prescribed as a boundary condition (Sect. 3.7). In the baseline simulations, we chose a layer depth of 150 hPa with respect to the surface pressure, corresponding to a layer depth of about 1.1 km at the standard pressure of 1013.25 hPa. It covers nine

pressure levels of the ERA-Interim and 19 levels of the ERA5 meteorological fields as prepared for use with the MPTRAC model (see Sect. 2.1).

## 3 Results

### 3.1 Examples of ECP and non-ECP particle dispersion simulations

Figure 3 and the video supplement of this paper (Hoffmann et al., 2023) illustrate the effects of the ECP on Lagrangian particle
dispersion simulations with MPTRAC using ERA5 and ERA-Interim meteorological fields. The specific model settings for the examples are described in Sect. 2.4. The non-ECP simulation with ERA-Interim does not show any resolved deep convective updrafts reaching the upper troposphere after 12 or 36 h of simulation time (Fig. 3a,b). In contrast, the non-ECP simulation with ERA5 reveals two resolved deep convective events over the Atlantic Ocean after 12 h and additional activity after 36 h of simulation time (Fig. 3c,d), where air parcels are directly injected into the tropical upper troposphere. While the ERA5 and
ERA-Interim non-ECP simulations show similar low level updrafts related to shallow convection (Fig. 3b,d), disagreement in the deep convective events relates to the limited spatiotemporal resolution and capabilities of ERA-Interim compared to ERA5 in resolving these events. The ECP simulation with ERA5 (Fig. 3e,f) and ERA-Interim (not shown) both much more rapidly populate the free troposphere in the selected region than the non-ECP simulations. The example illustrates how the ECP yields, even over a limited range of CAPE values, quite intense vertical mixing of air masses within the convective columns.

### 3.2 Statistics of explicitly resolved and parametrized convective updrafts

In this section, we discuss the statistical distributions of explicitly resolved and parametrized convective updrafts of ERA5 and ERA-Interim, applying the model settings and test configuration outlined in Sect. 2.4. Figures 4 and 5 show comparisons of zonal mean probability density functions (PDFs) of potential temperature change $\Delta\theta$ per 6 h for various non-ECP and ECP trajectory calculations, respectively. Note that we calculated the zonal PDFs using the initial latitudes of the trajectory seeds.
The final latitudes of the trajectories were not considered as the actual latitudinal displacement of the trajectories within the 6 h time range is rather small compared to the bin size. While Figs. 4 and 5 show the zonal distributions of the convective updrafts, Fig. 6 shows the globally averaged occurrence frequencies, allowing for a better quantitative comparison of the total numbers of events.

In general, the zonal PDFs of the non-ECP trajectory calculations in Fig. 4 reveal the strongest updrafts with about 60 K
per 6 h due to convective activity in the vicinity of the ITCZ. Beyond weaker convection and more prominent downwelling in the subtropics, secondary maxima of updrafts are found at middle latitudes. Updrafts in the polar regions are mostly below 10 – 15 K per 6 h in Northern Hemisphere polar summer and below 20 – 25 K per 6 h in Southern Hemisphere polar winter. Overall, the features of the zonal PDFs found here are expected and stress the important role of global circulation patterns such as the tropical Hadley cell and mid-latitude storm tracks in affecting the formation and occurrence of convection (Oort and
Yienger, 1996; Diaz and Bradley, 2004).

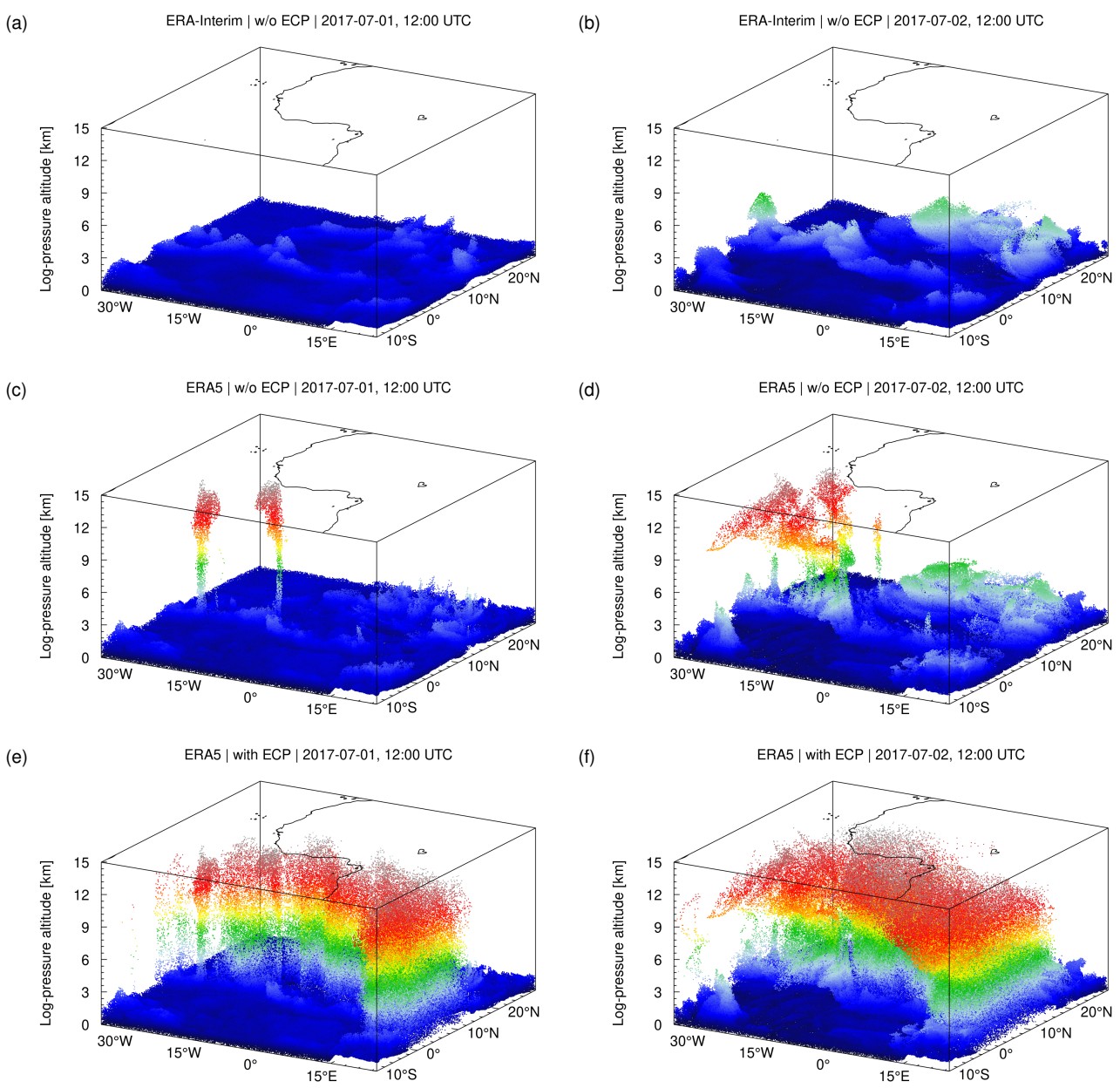

**Figure 3.** Three-dimensional air parcel distributions of non-ECP ERA-Interim (a,b), non-ECP ERA5 (c,d), and ECP ERA5 (e,f) Lagrangian particle dispersion simulations with MPTRAC. Particles were launched in the model lower boundary layer on 1 July 2017, 00:00 UTC. Air parcel distributions are shown after 12 h (a,c,e) and 36 h (b,d,f) of simulation time. ECP simulations were conducted with a threshold of $CAPE_0 = 1000\,J\,kg^{-1}$, including parametrized moderate to strong convective events, and 3-hourly event frequency. The color coding shows the log-pressure height of the air parcels.

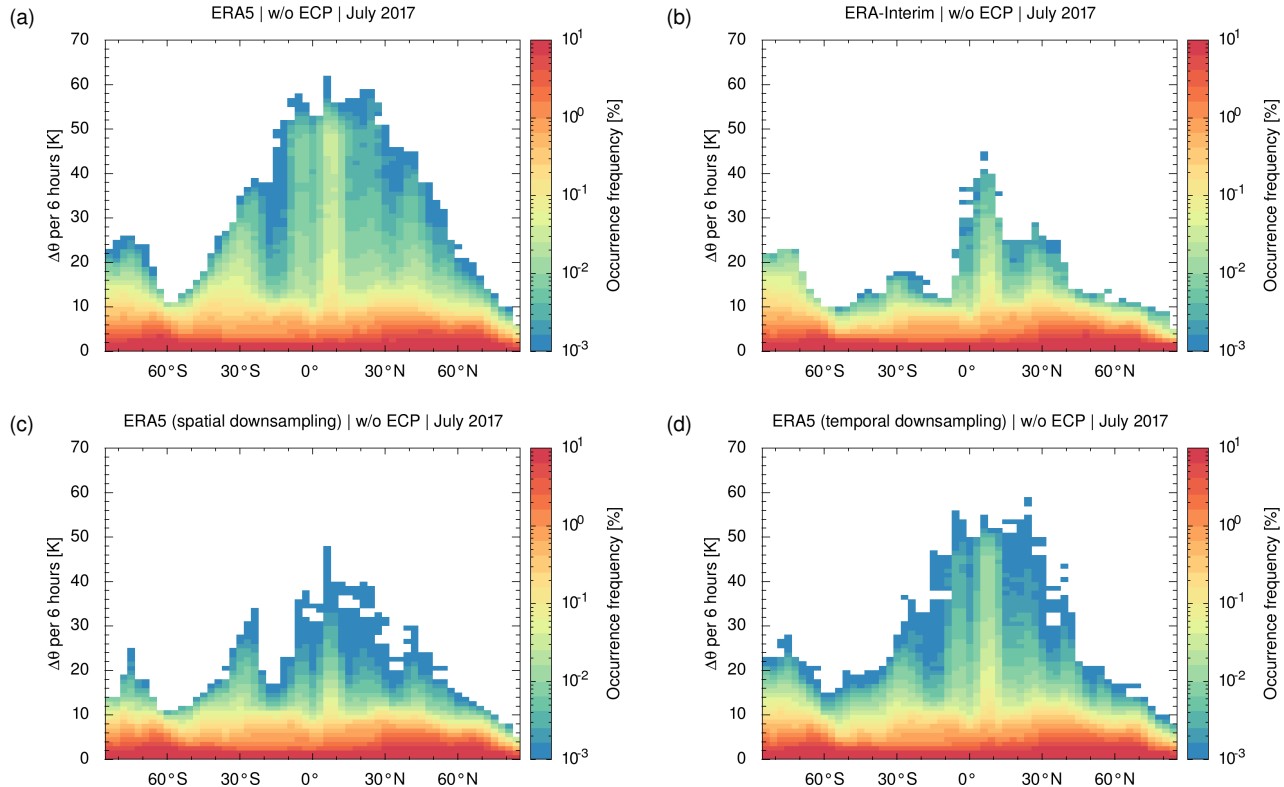

**Figure 4.** Zonal PDFs of explicitly resolved convective updrafts as represented by potential temperature change per 6 h time intervals along non-ECP trajectories for July 2017. Color bar ranges have been restricted to highlight strong updrafts. Results are shown for (a) ERA5 at full resolution, (b) ERA-Interim as well as (c) spatially and (d) temporally downsampled ERA5 fields. PDFs are calculated with bin sizes of $3°$ in latitude and $1\,\mathrm{K}/(6\,\mathrm{h})$ in potential temperature change.

Comparing the statistics of the non-ECP trajectories, we found that ERA5 (Fig. 4a) shows stronger and more frequent updrafts than ERA-Interim (Fig. 4b). This is consistent with earlier work (Hoffmann et al., 2019), demonstrating that ERA5 better resolves convective features due to improved spatiotemporal resolution of the ECMWF forecasting system. For the ERA5 reanalysis, we conducted two additional non-ECP calculations in which we downsampled the ERA5 fields from full 340 temporal resolution to 6-hourly time intervals and in which reduced the spatial resolution with a downsampling factor of $3 \times 3$ in the horizontal domain and a downsampling factor of two in the vertical domain. These downsampling factors where chosen to achieve a temporal or spatial resolution that is roughly comparable to ERA-Interim. The methodology of downsampling applied here is described in more detail by Hoffmann et al. (2019). The analysis of the downsampled ERA5 fields suggests that spatial resolution (Fig. 4c) is more relevant than temporal resolution (Fig. 4d) in maintaining the explicitly resolved updrafts, 345 as spatially downsampled ERA5 fields show significantly less peak updrafts than temporally downsampled data. With spatial downsampling being applied, the ERA5 updraft statistics become rather similar to the lower resolution ERA-Interim results.

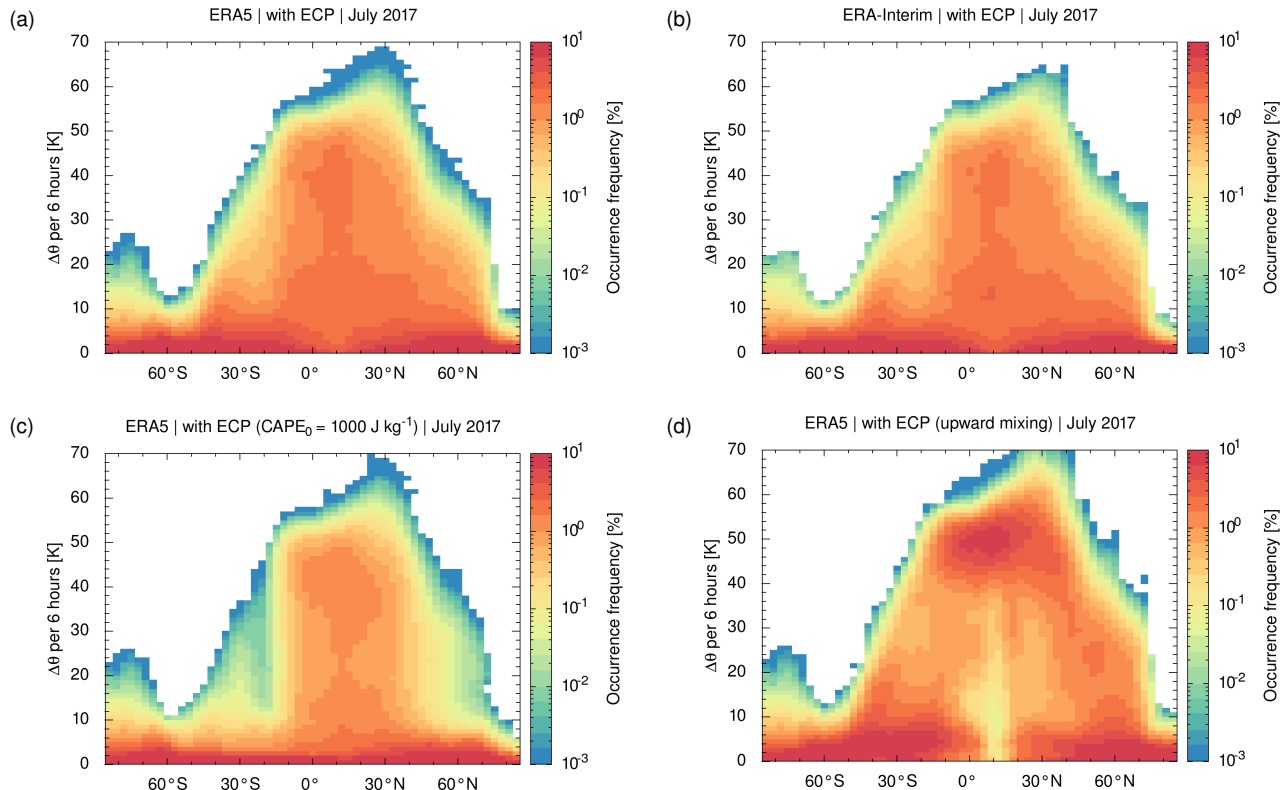

**Figure 5.** Same as Fig. 4, but for ECP trajectory calculations including parametrized updrafts. Results are shown for (a) ERA5 with an ECP threshold of $CAPE_0 = 0$, (b) ERA-Interim with $CAPE_0 = 0$, (c) ERA5 with a modified ECP threshold of $CAPE_0 = 1000\,J\,kg^{-1}$, and (d) ERA5 with $CAPE_0 = 0$ and with the ECP method being restricted to upward mixing.

Comparing the statistics of non-ECP (Fig. 4a,b) and ECP (Fig. 5a,b) calculations for ERA5 and ERA-Interim, it becomes obvious that the ECP substantially increases the number of convective updrafts in the tropics and at middle latitudes. The occurrence frequency of vertical updrafts in the range of 20 to 60 K per 6 h increases by up to 3 orders of magnitudes for
ERA5 and up to 5 orders of magnitude for ERA-Interim from the none-ECP to the ECP cases (Fig. 6). Despite the fact that the statistics are calculated from different meteorological fields, the ECP statistics of ERA5 (Fig. 5a) and ERA-Interim (Fig. 5b) are quite similar. This is promising, as we would expect the ECP to yield similar effects, independent of the different input data. Also note that the ECP and non-ECP patterns found here are qualitatively similar to the updraft statistics of Konopka et al. (2022, Fig. 7), despite the fact that their study used a different parametrization of convective uplifts implemented in another
Lagrangian transport model.

For the parametrized updrafts in Fig. 5, we mostly applied the ECP with a threshold of $CAPE_0 = 0$ at each time step of the model ($\Delta t = 180\,s$), thereby focusing on the extreme convection case, where parametrized convection takes places everywhere where CAPE is present. Figure 5c shows how the updraft statistics are changing when the threshold is set to

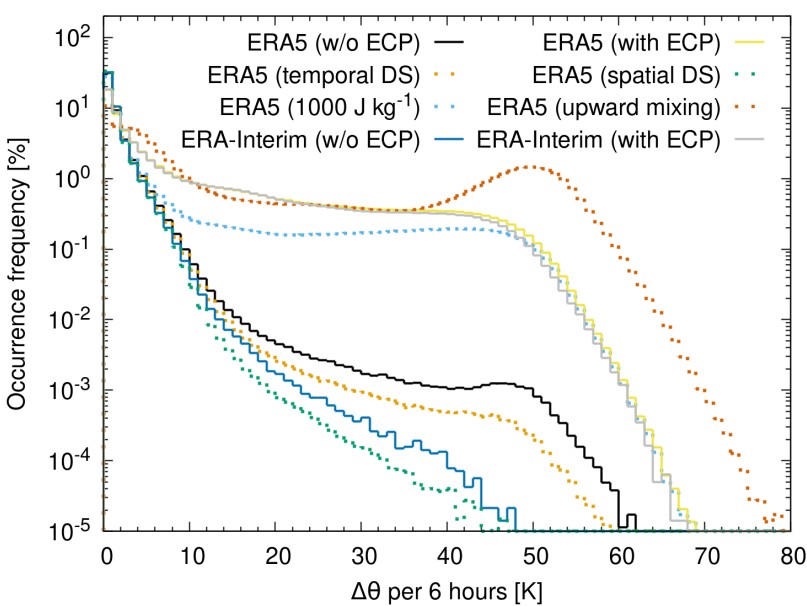

**Figure 6.** Globally averaged cumulative occurrence frequencies of the convective updrafts in Figs. 4 and 5.

$CAPE_0 = 1000 \, J \, kg^{-1}$. This test stresses the important role of the strong convective events on the parametrized convection, as
filtering the weak to moderate events with the increased threshold has only minor impact of the updraft statistics.

In the literature discussing the ECP and similar convection parametrizations for Lagrangian transport models, there is some ambiguity about whether vertical mixing in the convective columns is restricted to being directed "upward" or not (Gerbig et al., 2003; Stein et al., 2015; Konopka et al., 2019, 2022; Loughner et al., 2021). Restricting the mixing to the upward direction is likely motivated by the fact that convective updrafts occur on smaller, unresolved horizontal scales compared to
the compensating effects of larger scale downdrafts and subsidence. We implemented an option in the MPTRAC model to enforce upward mixing, i. e., at each convective step, the vertical displacement due to the mixing can only be positive. In this case, upward mixing is still being weighted by density to fulfill the well-mixed criterion. Figure 5d shows that upward mixing leads to more frequent and even stronger updrafts than the regular ECP method. Figure 6 shows that peak potential temperature changes increase by another 10 K per 6 h. However, note that the upward mixing approach requires tuning and a well-informed
choice of the time interval at which parametrized convection is being applied. Applying the upward mixing at each time step of the model, most air parcel will eventually be uplifted closely towards the EL. As the ECP with upward mixing will overestimate upward transport without tuning the convective event frequency, this approach was not further assessed in this study.

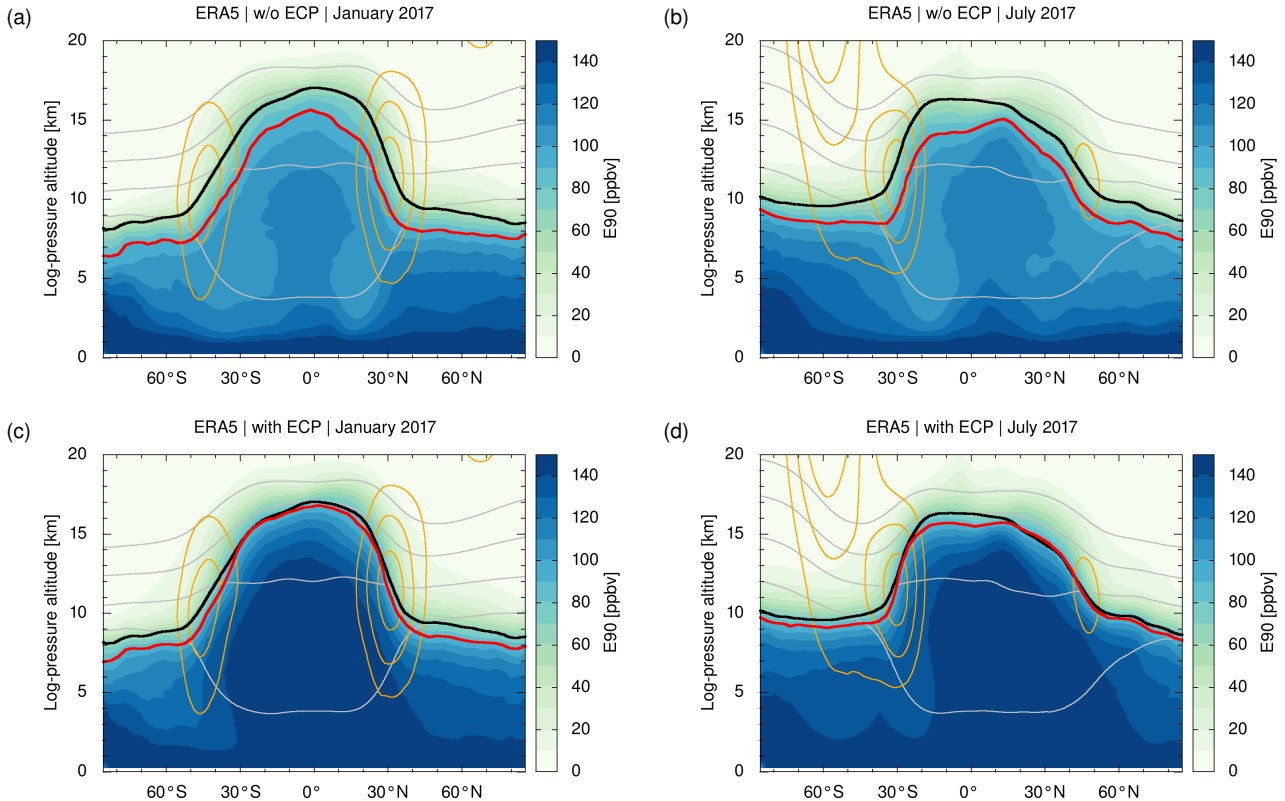

**Figure 7.** Artificial tracer e90 monthly mean zonal means (contour surface) in (a,c) January and (b,d) July 2017. Lagrangian transport simulations (a,b) without ECP or (c,d) with ECP were driven by ERA5. The red curve indicates the 90 ppbv contour of e90. The black curve indicates the dynamical tropopause (see text for details). Gray contours indicate potential temperatures of 320, 350, 380, and 410 K (from bottom to top). Orange contours indicate zonal winds at levels of 20, 30, and 40 m s$^{-1}$.

### 3.3 Comparison of e90 artificial tracer ECP and non-ECP simulations

The statistical assessment of the convective updrafts presented in Sect. 3.2 suggests large potential impact of the ECP on Lagrangian transport simulations for the troposphere for both, ERA5 and ERA-Interim. In the following sections, we discuss transport simulations of the artificial tracer e90 to quantify the impact of the ECP on the tracer transport using the model settings and test configurations described in Sect. 2.4. Figure 7 shows the e90 monthly mean zonal means for January and July 2017 of the non-ECP and ECP simulations driven by ERA5. In the figure, the 90 ppbv contour line of e90 has been highlighted to allow for comparison with the zonal mean monthly mean tropopause. For reference, we here show the dynamical tropopause, based on thresholds of 380 K of potential temperature in the tropics and 3.5 potential vorticity units (PVUs) in the extratropics (Hoffmann and Spang, 2022). Compared to the standard thermal lapse rate tropopause, the dynamical tropopause is better defined for polar winter isothermal temperature conditions (Zängl and Hoinka, 2001). Potential temperature contours in Fig. 7 indicate the stratification of the atmosphere. Zonal wind contours show the locations of the subtropical and polar jets.

For the non-ECP simulations, it is found that e90 concentrations gradually decrease with height from the surface towards the tropopause (Fig. 7a,b). Local maxima of e90 in the middle and upper troposphere are found in the tropics and local minima are found in the subtropics. The 90 ppbv contour of e90 resembles the shape of the dynamical tropopause but underestimates its height by $1-2$ km. In general, the zonal mean distributions found here are rather similar to results presented in other studies, for example the climatology of e90 concentrations from a Whole Atmosphere Community Climate Model (WACCM) run of Abalos et al. (2017) or the e90 tracer simulations with the Chemical Lagrangian Model of the Stratosphere (CLaMS) of Konopka et al. (2019, 2022). This indicates that the MPTRAC model yields a reasonable representation of tracer transport in the free troposphere and stratosphere in the present simulation set up.

In contrast to the non-ECP simulations, the ECP simulations with MPTRAC led to significantly larger e90 concentrations in the free troposphere (Fig. 7c,d). For instance, the middle and upper troposphere e90 maxima in the tropics ($30°$S to $30°$N) were increased from $110-120$ ppbv to $140-150$ ppbv when using the ECP. The 90 ppbv contour of e90 now even more closely resembles the dynamical tropopause, with height differences well below $\pm1$ km. The comparison of the non-ECP and ECP simulation results indicates that unresolved, parametrized convection has strong impact of tracer transport in the free troposphere, in particular at tropical latitudes, which are governed by frequent and intense convective activity. This shows that even state-of-the-art reanalyses such as ERA5 with much improved spatiotemporal resolution compared to earlier reanalyses require a convection parametrization in Lagrangian transport models to properly represent transport from the planetary boundary layer into the free troposphere.

### 3.4 Comparison of e90 artificial tracer simulations driven by ERA5 and ERA-Interim

To assess the influence of the meteorological input data on the Lagrangian transport simulations, we conducted the e90 transport simulations with ERA-Interim instead of ERA5. The differences of the e90 monthly mean zonal means of ERA5 minus ERA-Interim are shown in Fig. 8, respectively. For the non-ECP simulations (Fig. 8a,b), it is found that the ERA-Interim simulation mostly underestimates the e90 concentrations in the free troposphere compared to ERA5. The underestimation becomes as large as 15 to 20 ppbv in the tropical upper troposphere. This underestimation is attributed to the fact that explicit updrafts are under-represented in ERA-Interim compared to ERA5 (see Sect. 3.2).

For the ECP simulations (Fig. 8c,d), there is generally better agreement between the ERA5 and ERA-Interim simulations than for the non-ECP simulations (Fig. 8a,b). This may be attributed to the fact that in the ECP simulations the e90 distributions are largely governed by parametrized updrafts, which exhibit statistically similar distributions between ERA5 and ERA-Interim (Sect. 3.2). The largest e90 differences between the ECP simulations are in the range of $\pm15$ ppbv and found at the tropopause. Above the tropical tropopause, e90 from ERA5 is lower than ERA-Interim, indicating slower transport in the tropical pipe in ERA5 than in ERA-Interim. This is consistent with recent studies on the Brewer–Dobson circulation finding that tropical upwelling in ERA5 is up to 40% weaker than in ERA-Interim, which is mainly due to significantly weaker gravity wave forcing at the equatorial-ward upper flank of the subtropical jet (Diallo et al., 2021; Ploeger et al., 2021). In contrast, ERA5 yields larger e90 concentrations than ERA-Interim at subtropical and middle latitudes, suggesting stronger isentropic mixing between the tropical upper troposphere and the extratropical lowermost stratosphere in ERA5.

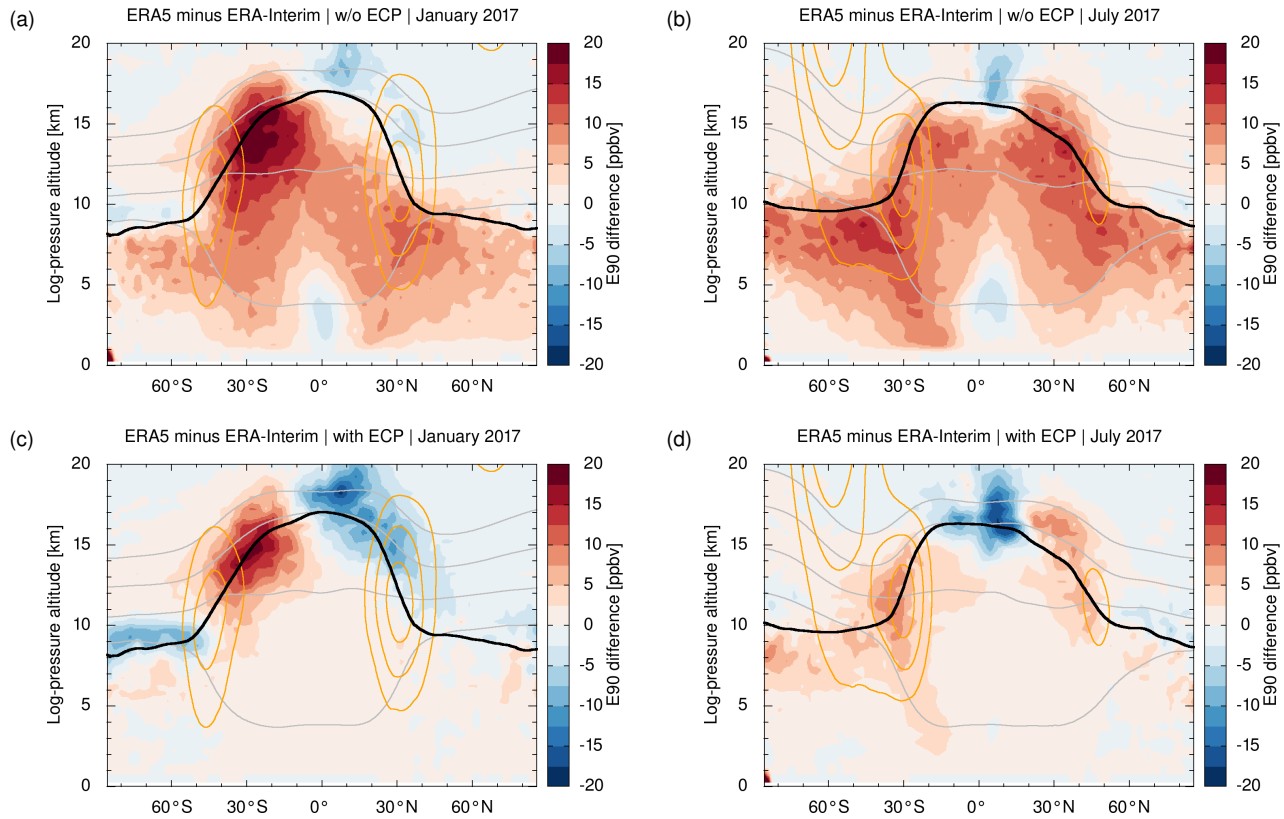

**Figure 8.** Differences of e90 monthly mean zonal means from (a,b) non-ECP and (c,d) ECP transport simulations driven by ERA5 and ERA-Interim for (a,c) January and (b,d) July 2017. Differences are shown on a range of $\pm 20$ ppbv, whereas e90 has a maximum volume mixing ratio of 150 ppbv per definition. See Fig. 7 for comparison.

### 3.5 Sensitivity of ECP simulations on the CAPE threshold

In this section, we discuss a sensitivity test on the threshold $CAPE_0$ used to trigger convective events in the ECP simulations. The model settings and test configuration are described in Sect. 2.4. Figure 9a shows the sensitivity of the 90 ppbv contour of e90 on the choice of $CAPE_0$. The test was conducted using ERA5 meteorological fields. We focus the discussion on July 2017, noting that other months show similar results. We tested CAPE threshold values in the range of 100 to 5000 J kg$^{-1}$. Except for minor differences, the simulation results for thresholds of 100 to 500 J kg$^{-1}$ are quite similar to the extreme case without any restrictions on CAPE. The 90 ppbv contour of e90 is located near the dynamical tropopause. For CAPE thresholds of 1000 and 2000 J kg$^{-1}$, restricting the ECP to moderate to strong convective instability, increasing differences in the 90 ppbv contour become visible in the extratropics. The CAPE threshold of 5000 J kg$^{-1}$ filters all events except for few local cases of extreme instability. For this threshold, the e90 contour matches the non-ECP simulation.

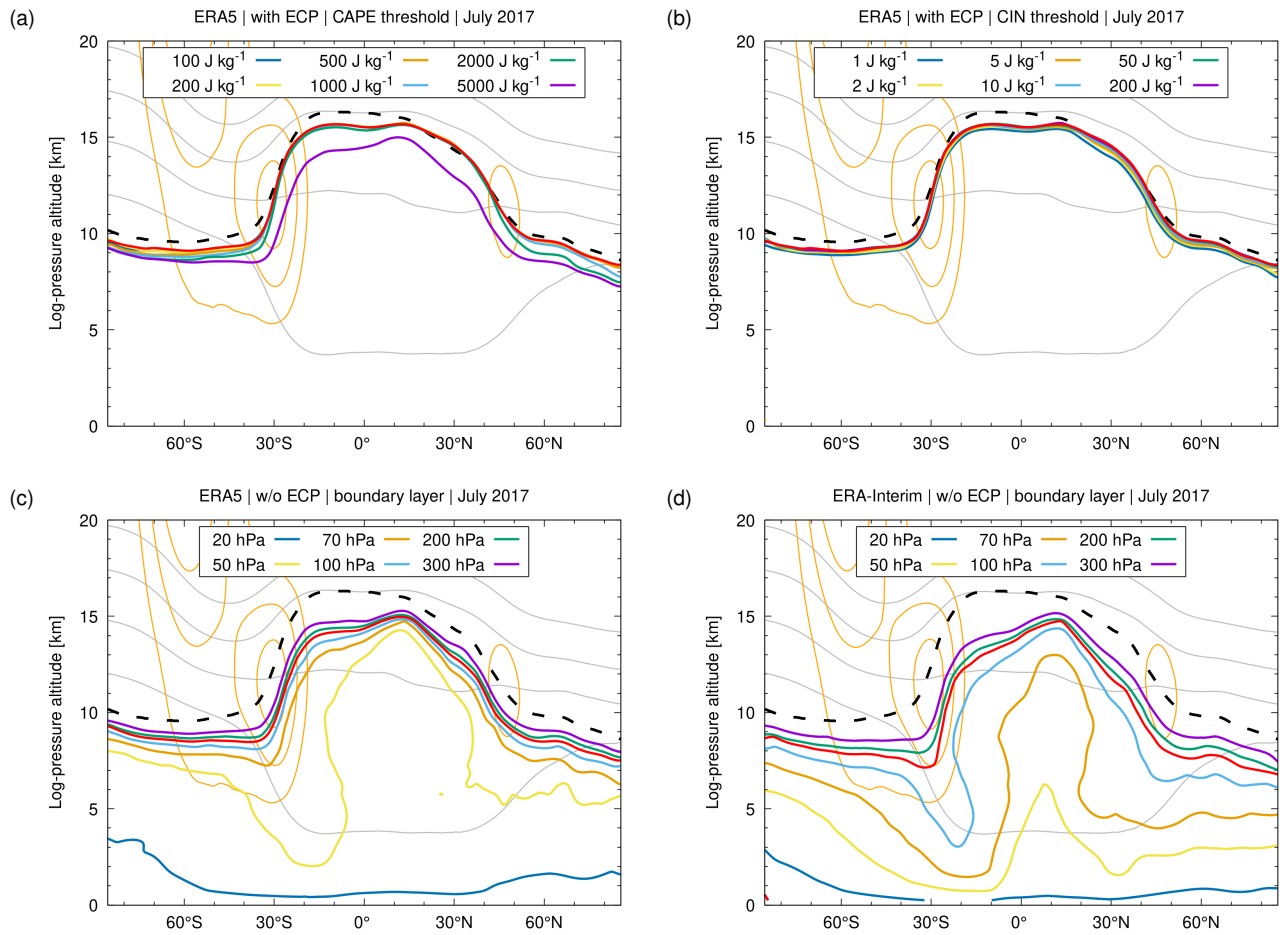

**Figure 9.** Sensitivity test on ERA5 transport simulations with respect to the CAPE threshold in ECP simulations (a), an additional CIN threshold in ECP simulations (b), and the surface layer depth in non-ECP simulations driven by ERA5 (c) and ERA-Interim (d). Shown are 90 ppbv contour lines of e90 monthly mean zonal means in July 2017 for different parameter settings (see plot titles). The red curves show the 90 ppbv contour for default settings of (a) $CAPE_0 = 0$, (b) no CIN filter, and (c) 150 hPa surface layer depth. The black dashed curve shows the dynamical tropopause. See Fig. 7 for further details.

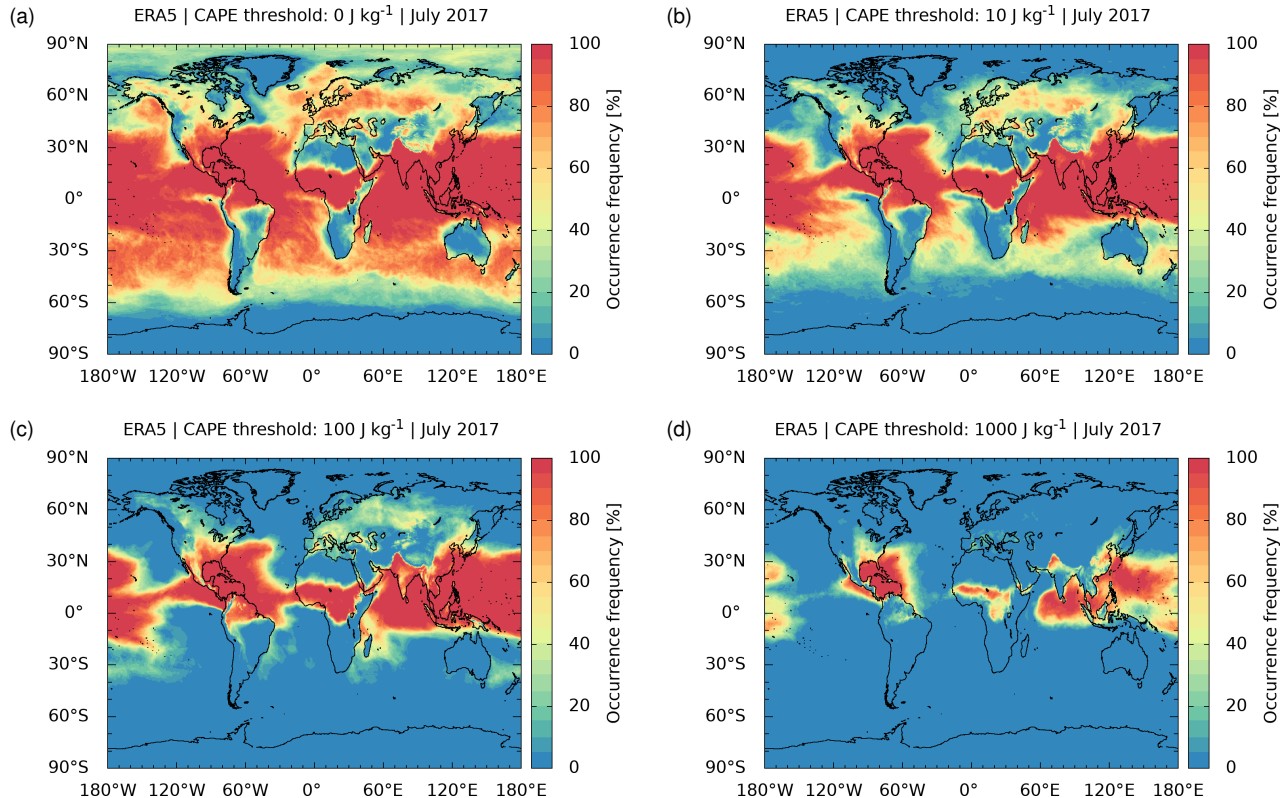

**Figure 10.** Monthly mean occurrence frequencies of ECP convective events in July 2017 for ERA5. Statistics are shown for different threshold values $CAPE_0$ triggering the events (see plot titles).

Figure 10 shows global maps of the occurrence frequencies of the ECP convective events for different thresholds $CAPE_0$ for July 2017 ERA5 meteorological fields. In general, the largest occurrence frequencies (up to 100 %) are found over the tropics, and the frequencies gradually decrease towards middle and high latitudes. The occurrence frequencies decrease notably with increasing $CAPE_0$, where $CAPE_0 = 0$ (Fig. 10a) includes all events where CAPE exists, whereas $CAPE_0 = 1000\,\mathrm{J\,kg^{-1}}$ (Fig. 10d) is putting the focus on moderate to strong convective events. With increasing $CAPE_0$, the simulated convective events are more localized on global hotspots of convection. The local maxima of the CAPE-based occurrence frequencies coincide with maxima of precipitation and deep convective clouds over the Gulf of Mexico and the Caribbean Sea, Central Africa and Western Africa, the Northern Indian Ocean, and the tropical Western Pacific as seen in satellite records (Liu et al., 2007; Spang et al., 2012).

The sensitivity test shows that the CAPE threshold is an important control parameter of the ECP simulations. Strong convective events with CAPE values of $2000\,\mathrm{J\,kg^{-1}}$ or more play a major role in affecting the zonal mean e90 distributions throughout the troposphere. Strong convective events are associated with mean ELs at log-pressure heights of 10 to 12 km or more for larger CAPE values (see Sect. 2.3). This includes numerous events of convective storms at middle and high latitudes

and deep convection in the tropics, where convective updrafts reach height levels close to the tropopause. Locally, the effects of choosing the CAPE threshold might be even more significant, as for instance tropical convection is focused on global hotspots which exhibit large intermittency and variability over space and time.

### 3.6 Sensitivity of ECP simulations on the CIN threshold

Figure 9b shows the results of a sensitivity test of the threshold $CIN_0$ on the e90 tracer transport simulations for the modified ECP method (Sect. 2.3). The model settings and test configuration are described in Sect. 2.4. We tested $CIN_0$ values of 1 to $200 \, J \, kg^{-1}$. Overall, the test reveals only weak sensitivity of the 90 ppbv contour of the e90 artificial tracer on $CIN_0$. The weak dependency on the parameter $CIN_0$ in this zonal mean view is attributed to the fact that the global e90 distributions in the free troposphere are mostly governed by the strong convective updrafts, which are generally not filtered and removed by a CIN threshold. The CIN threshold is therefore expected to only locally affect the e90 distributions.

Figure 11 shows maps of occurrence frequency differences of the ECP convective events for different thresholds $CIN_0$ minus unfiltered data (Fig. 10a). This analysis shows that small CIN thresholds significantly reduce the occurrence of parametrized convective events on the global scale whereas large CIN thresholds have only local effects. For a CIN threshold of $2 \, J \, kg^{-1}$, the global event frequencies are reduced up to 50 to 100 % over ocean and land. For a CIN threshold of $200 \, J \, kg^{-1}$, the spatial patterns of the convective events mostly resemble the unfiltered case, except for specific regions (i. e., North American Great Plains, Mediterranean Sea, Central Africa and Western Africa, Arabian Sea), where CIN shows local maxima and the CIN filter therefore still has strong effects. A recent study of Clemens et al. (2023) shows how the additional CIN threshold in the ECP can be used to remove unrealistic parametrized convection events over the Persian Gulf in an August 2016 case study investigating source regions of the Asian Tropopause Aerosol Layer (ATAL) on the Indian subcontinent.

### 3.7 Sensitivity on the depth of the surface layer

The results of the sensitivity test for the layer depth for the non-ECP simulations are presented in Fig. 9c,d for ERA5 and ERA-Interim, respectively. The test reveals a strong dependence of the simulated e90 distributions on the depth of the surface layer. For ERA5, the simulated 90 ppbv contours largely deviate from the baseline simulation for layer depths of 20 hPa (about 140 m) to 70 hPa (about 500 m). For ERA-Interim, there is even stronger dependence on the layer depth, such that even the 100 hPa (about 730 m) case notably differs from the baseline simulation. Note that even for larger layer depths of 200 hPa (1.5 km) and 300 hPa (2.5 km) the 90 ppbv contour of e90 still remains about 1 to 2 km below the dynamical tropopause, suggesting that transport into the upper troposphere is underestimated compared to the ECP case. This test indicates that significant updrafts are being present in the ERA5 reanalysis. However, the updrafts are not extending down to the surface and are therefore not be captured in the non-ECP simulations, if the selected surface layer is too thin. The sensitivity test for the non-ECP simulations indicates that estimates of convective mass flux from a near-surface layer into the free troposphere will strongly depend on the depth of the layer.

In contrast, for the ECP simulations, the sensitivity test did not reveal any significant variations in the e90 concentrations with respect to the surface layer (not shown). In the ECP simulations, the e90 concentrations in the free troposphere are largely

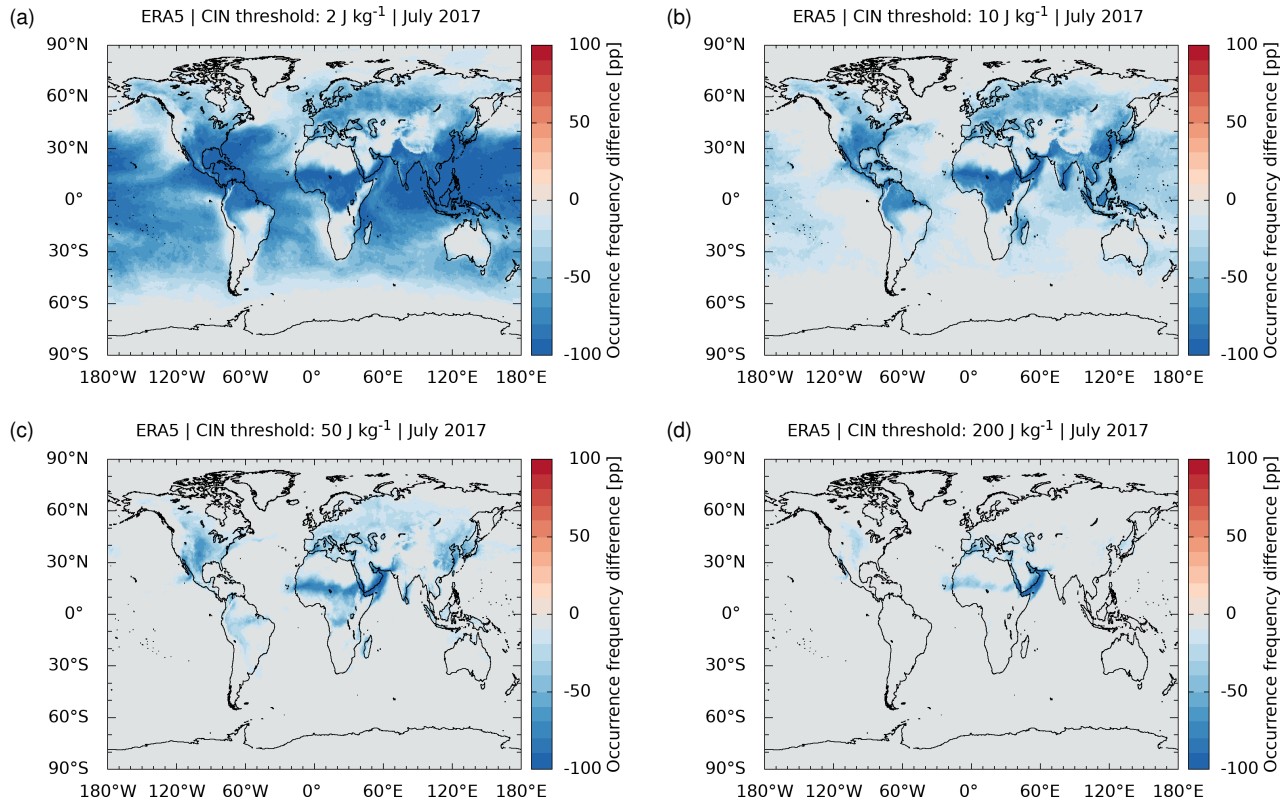

**Figure 11.** Differences of filtered minus unfiltered monthly mean occurrence frequencies of ECP convective events in July 2017 for ERA5. Statistics are shown for different threshold values $CIN_0$ used for suppressing convective events (see plot titles).

governed by the parametrized rather than the explicitly resolved updrafts of the reanalyses. Following other studies (Gerbig et al., 2003), the ECP was implemented here to influence all parcels in the convective columns down to the surface. In principle, the lower boundary of the ECP could be changed to other levels, for example, the surface pressure could be replaced by the level of free convection. This might be considered physically more realistic, but it would cause other difficulties as the MPTRAC model does not feature any advanced parametrizations for turbulence and mixing in the planetary boundary layer. Changing the lower level of the convective columns in the ECP scheme might therefore cause similar issues as seen in the non-ECP tests, preventing air parcels from being captured by convective updrafts, if the chosen surface layer is too thin. For this reason, we follow the original ECP approach and apply convective mixing down to the surface.

## 4 Conclusions

In this study, we assessed the impact of the ECP on Lagrangian transport simulations for the free troposphere and the lower stratosphere. The ECP is conceptually simple and computationally fast. It requires only two input variables from the meteorological fields, CAPE and EL. If CAPE exceeds a given threshold $CAPE_0$, a parametrized convective event is triggered, i. e.,

air parcels are vertically mixed in the convective column from the surface to the EL. If there is already an explicitly resolved convective updraft being present in the meteorological fields at the same location, this would be no harm, as the ECP would simply override this explicit updraft with a parametrized event. An important aspect is that the vertical redistribution of the air parcels is being weighted by air density, ensuring that the well-mixed criterion is being fulfilled. Next to the standard ap-

490 proach, we propose a modification of the ECP by considering CIN to suppress convection events where stable layers are likely preventing convection in the real atmosphere. For our assessment, we implemented the ECP and the proposed extension into the Lagrangian transport model MPTRAC.

For the month of July 2017, we conducted a statistical analysis of both, explicitly resolved and parametrized convective updrafts, by analyzing potential temperature change along 6-hour trajectories calculated with the ECMWF reanalyses. Similar

to Hoffmann et al. (2019), the statistical analysis for the non-ECP case showed that ERA5 explicitly resolves peak updrafts more frequently than ERA-Interim. Spatiotemporal downsampling of ERA5 to the resolution of ERA-Interim reduces the frequency of the strong updrafts, making the statistics comparable to ERA-Interim. This stresses the important role of the resolution of the meteorological fields on explicitly resolving convection. However, although ERA5 already improves upon ERA-Interim, we found that trajectory calculations using the ECP still much further increase the number of peak updrafts, i. e.,

by 2 to 3 orders of magnitude for ERA5 and 3 to 5 orders of magnitude for ERA-Interim. Similar findings have been reported by Konopka et al. (2022), using a different Lagrangian transport model and a different convection parametrization scheme. Overall, this analysis indicates large potential impact of applying convection parametrization schemes such as the ECP on Lagrangian transport simulations for the troposphere driven by global, coarse-resolution meteorological fields from reanalyses or forecasts.

To assess the impact of the ECP on atmospheric transport, we conducted e90 artificial tracer simulations driven by ECMWF's ERA5 and ERA-Interim reanalyses. The comparison reveals large differences between ECP and non-ECP simulations. Without the ECP, e90 in monthly mean zonal mean distributions was underestimated compared to the ECP simulations. The underestimation was more severe for ERA-Interim than for ERA5. With the ECP being applied, we found more realistic distributions of e90. In particular, the 90 ppbv contour of e90 more closely resembled the dynamical tropopause in the ECP case, which is

expected as this contour is meant to represent the chemical tropopause (Prather et al., 2011; Abalos et al., 2017). Differences of ECP simulations between ERA5 and ERA-Interim are smaller than differences between the non-ECP simulations, indicating that the ECP simulations are largely dominated by the parametrized convective updrafts, which have similar characteristics in ERA5 and ERA-Interim.

We performed several sensitivity tests on the ECP simulations. A test of the $CAPE_0$ threshold used to trigger parametrized

convective events showed that a threshold of $5000 \, J \, kg^{-1}$ removes most events except for a few local cases of extreme instability. For such large thresholds, the zonal mean 90 ppbv contour of the artificial tracer e90 agrees with the non-ECP simulation. CAPE thresholds of 1000 to $2000 \, J \, kg^{-1}$ restrict the ECP to cases of moderate to strong convective instability, resulting in small differences in the zonal mean 90 ppbv contour of artificial tracer e90 compared to the extreme case without CAPE restrictions. For smaller CAPE thresholds, we found no significant differences in the zonal mean contour with respect to $CAPE_0 = 0$. The

sensitivity test therefore showed that, from a global perspective, the simulated transport is mostly affected by strong to extreme

convective events. CAPE values larger than $1000\,\mathrm{J\,kg^{-1}}$ are associated with mean EL log-pressure heights of 10 to $12\,\mathrm{km}$, indicating that parametrized deep convective updrafts in the tropics have the strongest influence on e90 distributions in the free troposphere.

Considering the proposed modification of the ECP method, we conclude that introducing the threshold $CIN_0$, which prevents the occurrence of parameterised convective updrafts for warm, stable layers, leads to local improvements in areas with climatically large CIN, e. g., over North Africa and the Arabian Peninsula. This aspect is further elaborated in a recent study by Clemens et al. (2023), which used Lagrangian transport simulations to identify the source regions of the Asian tropopause aerosol layer over the Indian subcontinent in August 2016. The study of Clemens et al. (2023) found that using a CIN threshold of $50\,\mathrm{J\,kg^{-1}}$ helps to remove spurious parameterised convection events over the Persian Gulf in the case of ECP simulations with MPTRAC. On a global scale, the modification of parametrized convective events via $CIN_0$ was found to have less impact on the zonal mean 90 ppbv contour of the e90 artificial tracer in this study.

Another test revealed the sensitivity of the simulation results on the depth of the near-surface layer used to prescribe the e90 boundary conditions. We found a strong dependency of the non-ECP simulations on the layer depth, whereas the ECP simulations were not affected. Explicit updrafts in the reanalyses occur at vertical levels well above the surface, whereas the ECP impacts all air parcels down to the surface. This is a complicated issue, as by definition the e90 tracer is released "at the surface". Unlike other Lagrangian particle dispersion models, MPTRAC does not provide distinct parametrizations of turbulence and mixing in the planetary boundary layer. Models with stronger turbulence and mixing in the boundary layer might show less dependence on the depth of the near-surface layer and generally yield more transport from the near-surface layer into the free troposphere. This issue might be further assessed in future work, for example by comparing transport simulations with MPTRAC with other models.

While this study provided initial guidance, future work on the ECP should focus on better tuning the parameters $CAPE_0$ and $CIN_0$ of the parametrization. Considering an artificial tracer such as e90 is helpful to assess the effects of the ECP on global transport simulations. However, parameter tuning should involve also real measurements of tropospheric tracers such as water vapor, carbon monoxide, etc. We are currently conducting two new studies assessing the effects of the ECP on Lagrangian transport simulations for the Asian tropopause aerosol layer (Clemens et al., 2023) and the Ambae Island volcanic eruption (Liu et al., 2023). Also, comparisons with more sophisticated convection parametrizations for Lagrangian models would be of interest (e. g., Brinkop and Jöckel, 2019; Konopka et al., 2019; Wohltmann et al., 2019; Konopka et al., 2022). Nevertheless, based on the present results, we conclude that Lagrangian transport simulations for the free troposphere driven by limited resolution, global reanalyses benefit from a convection parametrization such as the ECP to more realistically represent transport from the planetary boundary layer into the free troposphere. This applies for ERA-Interim, but also for ERA5, despite its much improved spatiotemporal resolution.

*Code and data availability.* The MPTRAC model (Hoffmann et al., 2016, 2022a) is made available under the terms and conditions of the GNU General Public License (GPL) version 3. The release version 2.4 of MPTRAC applied in this paper has been archived on Zenodo

(Hoffmann et al., 2022b). Newer versions of MPTRAC are made available via the repository at https://github.com/slcs-jsc/mptrac (last access: 4 January 2023). The ERA5 and ERA-Interim reanalyses (Dee et al., 2011; Hersbach et al., 2020) were retrieved from ECMWF's Meteorological Archival and Retrieval System (MARS). See https://www.ecmwf.int/en/forecasts/datasets/browse-reanalysis-datasets (last access: 4 January 2023) for further details.

*Author contributions.* LH and PK jointly developed the concept for this study. LH implemented the ECP in MPTRAC and conducted the simulations and data analysis. JC and BV contributed to the sensitivity tests and modifications of the ECP method. All authors contributed to the interpretation of the results and writing the manuscript.

*Competing interests.* The authors declare that no competing interests are present.

*Acknowledgements.* ERA5 data were generated using Copernicus Climate Change Service Information. Neither the European Commission nor ECMWF are responsible for any use that may be made of the Copernicus information or data in this publication. We acknowledge the Jülich Supercomputing Centre for providing computing time and storage resources on the supercomputer JUWELS. We acknowledge Gebhard Günther and Olaf Stein for provisioning the ECMWF ERA5 and ERA-Interim reanalyses in Jülich. We thank our colleagues at the Institute of Energy and Climate Research and at the Jülich Supercomputing Centre for providing helpful feedback and suggestions on this study.

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
