# Peer review of "Lagrangian transport simulations using the extreme convection parametrization: an assessment for the ECMWF reanalyses"

_EGUsphere, 2023_

## Referee Comment (RC2)

Review of egusphere-2023-72

*'Lagrangian transport simulations using the extreme convection parametrization: an assessment for the ECMWF reanalyses"*

by Lars Hoffmann et al.

Paper in review in Atmospheric Chemistry and Physics

**1 General comment**

This study quantifies convective mass and tracer transport using the Lagrangian MPTRAC model driven by re-analysis data and coupled to the extreme convective parameterization ECP. The ECP randomly redistributes air parcels and tracers between the surface and the equilibrium level if a certain CAPE threshold is exceeded. The authors quantify 'explicit' versus parametrized convective updrafts and evaluate the choice of the required CAPE threshold to initiate convective transport. Moreover, the sensitivity of artifical tracer distribution to different settings of CAPE, CIN, vertical mixing and surface layer depth is evaluated.

The manuscript is well-structured, comprehensive and well-written. My major concern addresses the stated key research question and conclusion. The key question (l. 62 ff) addresses the question whether ERA5 (compared to ERA-Interim) is sufficient to properly represent convective updrafts and global tracer transport. My concern is that per definition, one would not expect ERA5 (nor ERA-Interim) to adequately represent convective features based on Lagrangian trajectory calculation from the resolved wind fields, as ERA5 uses a convection parameterization and does not represent convective updrafts explicitly. Thus, it is not surprising that ERA5 without ECP does not 'explicitly' resolve a lot of convective features, and requires a parameterization to efficiently vertically mix air parcels and tracers. Thus, the key conclusion (e.g., l. 18 ff and 466 ff) that ERA5 still needs a convection parametrization to better represent convective and tracer transport is expected given the properties of the re-analysis data. Moreover, it is expected that the spatially and temporally coarser ERA-Interim represents less convective features, which has also been already stated by Hoffman et al. 2019 (l. 41 ff and 422 ff). I think the manuscript could profit substantially from focusing on the sensitivity tests and idealised tracer experiments, the effects of implementing CIN, and more directly setting their results into perspective with other convective transport models, instead of comparing ERA5 and ERA-Interim without ECP, as well as ERA5 with and without ECP, which for the reasons mentioned provides expected results. Instead, I would suggest to streamline the manuscript focusing on sensitivity experiments and systematically evaluating their impact on global tracer distribution. Furthermore, the sensitivity tests and the influence of the tuning parameters CAPE and CIN on convective event frequency are shown and discussed. The results are somewhat expected: For example, if the threshold of CAPE to trigger convective mixing is increased, the frequency of convection is reduced in regions

that are climatologically characterized by lower CAPE values. Yet, it remains unclear to me which of the tested ECP setting is more realistic beyond the expected impact. Concerning this aspect, the study may benefit from setting their results into perspective with other studies, observations and comparison to more sophisticated convective parameterizations.

In summary, I think the manuscript (including abstract, conclusion and results) could benefit from focusing on key aspects of their sensitivity studies and streamlining the sensitivity experiments and novel key results. I have several additional questions and remarks to the authors that are listed below.

**Major comments**

1 Introduction
Although the introduction reads well, I find the content is rather general and recent studies and developments are not sufficiently explained. For example, l. 54 ff state that "various techniques and parametrizations have been developed to better represent the effects of convection in Lagrangian transport simulations". However, no details are provided which makes it difficult to understand the novelty of this study and to set results into perspective. I would ask the authors to provide a more in-depth overview of current research (which is to some extent refered to later in the manuscript).

2 Methods
I would ask the authors to include the first paragraph concerning the sensitivity to the depth of the surface layer into the Methods section (first paragraph in Section 3.7). Moreover, the different settings of the MPTRAC simulations (Sections 3.1, 3.2 as well as 3.3 l. 270-296) could also be placed in a subsection of the Methods. This study uses different settings for the ECP simulations (e.g., CAPE, CIN, vertical mixing, boundary layer depth), which could already be briefly introduced in the Methods part. I was wondering why the authors chose different settings for trajectory seeding? Does seeding on a regular global 0.3x0.3 grid introduce biases in convective mass transport as the seeds are not equidistant (for the tracer simulations a latitude weight was applied)? For convective transport the 0.3x0.3 grid was also applied to the coarser ERA-Interim data. In the case of no ECP parameterization (i.e. no random vertical re-distribution), does this result in trajectories that are quasi-identical in ERA-Interim?

3 Comparison of ERA-Interim and ERA5
I think the authors could focus on the sensitivity tests with different parameters / threshold for the ECP, instead of comparing ERA5 to ERA-Interim without ECP. It is not surpring that convective updrafts occur less frequently in the coarser ERA-Interim data. Moreover, as also mentioned in l. 41 ff and 422 ff Hoffmann et al. (2019) already showed that "ERA5 explicitly resolves peak updrafts more frequently than ERA-Interim". Moreover, trajectory calculations are hardly comparable due to the different spatiotemporal resolution, in particular, the coarse temporal

resolution of ERA-Interim of 6 h. Is it correct that ERA-Interim trajectories are calculated from only two timesteps (as 6-h trajectories are used)? I would thus suggest to focus on the comparison of different ECP settings, and how this affects global tracer distribution.

4 Results
In addition to the comments above, I believe that the Results Section could be improved by streamlining and rearranging some paragraphs. Please see also specific comments below.

**2 Specific comments and technical corrections**

1. l. 24 ff: Please streamline, and include a few relevant references (e.g., for convective mass transport into the stratosphere).

2. l. 30: The statement to shallow convection could be removed as it is not relevant in this study.

3. l. 32: Instead of referring to textbooks, the authors could include references about the convection parameterization that is applied in ERA5 and ERA-Interim.

4. l. 48 ff: This paragraph outlines that coarser re-analysis data sets are not capable to explicitly resolve small-scale convective features for various reasons. My question is why the authors use ERA5 and ERA-Interim without ECP as it is expected that convection is underestimated in both data sets? Instead, I would suggest to focus on the ECP settings required to obtain reasonable results. See also general comments.

5. l. 99 f: Please add the number of resulting pressure levels and the approximate vertical spacing in the lower troposphere where trajectories are seeded.

6. l. 118: 'particle data': Does this refer to air parcels? Please clarify.

7. l. 124 f: How was the timescale of 6 h as typical convective timescale determined? Please clarify and/or add references.

8. Figure 2a and l. 175 ff: Does the occurrence frequency show convective events and/or the frequency distributions of CAPE values or is this the same, since convective events are triggered simply by the CAPE threshold. Please clarify.

9. l. 177 ff: Given equation 1, a relation between CAPE and the height of the EL is given. Regarding the still considerable scatter in Fig. 2b, I'm not sure if CAPE only is a reliable quantity to estimate EL (e.g., for a CAPE value of $100 \, \mathrm{J \, kg^{-1}}$ the EL height varies between 3 and 10 km).

10. l. 196: Why did the authors seed trajectories at random vertical positions in the boundary layer and not at various fixed heights?

11. l. 200: How was the time interval for ECP of 3 h determined? For subsequent analysis, I believe a threshold of 180 s was used (l. 152). Does this apply to all systematic ECP simulations? How sensitive are simulations to this threshold? I would appreciate if this was discussed in the Methods Section.

12. l. 235: Could the authors please elaborate on the spatial downsampling? Are wind fields averaged, smoothed, etc. ?

13. l. 248-250: Could the authors please elaborate on the similarities and differences between this study and Konopka et al. (2022).

14. Figure 5: I would appreciate if the figures could consistently include labels for the applied CAPE threshold (e.g., as panel 5b).

15. Figure 6: I apologize if this pertains to our printer, but I can hardly distinguish the colors for (i) ERA5 (1000 J kg$^{-1}$) and ERA5 (spatial DS) as well as (ii) ERA5 (w/o ECP) and ERA-Interim (with ECP).

16. l. 270-296: Did the authors consider to move this paragraph to the Methods Section? See general comment 2.

17. l. 256 ff: Please cite the relevant literature here.

18. l. 256 ff: Comment related to vertical mixing: To streamline the manuscript, I believe the effects of vertical mixing could be removed. In particular, as no detailed information on potential tuning and the choice of time interval is provided and as it is not applied afterwards.

19. l. 265 and 354: Please consistently replace 'equilibrium level' by 'EL'.

20. l. 300: Please rephrase 'vertical layering' (e.g., stratification).

21. l. 315 f: Given the property of the underlying wind fields, this general statement is expected.

22. Figures 7 and 8: I think it could be sufficient to show January and July (e.g., move April and October to Appendix / Supplement) and show differences between Fig. 7 and 8 (similar to Fig. 9 showing differences between ERA5 and ERA-Interim), as all sub-panels look very similar.

23. l. 338: 'other months show similar results': They could be added to an Appendix (see also comment above).

24. l. 344 ff: Based on the global distribution of CAPE it is expected that the frequency of convective events decreases with increasing CAPE threshold. Please streamline this paragraph and/or move it to Section 3.2 (Statistics of explicitly resolved and parametrized convective updrafts).

25. l. 352 f: 'Moderate to strong convective events [...] play a major role in affecting the zonal mean e90 distributions throughout the troposphere': Based on Fig. 11a, I disagree with this statement. The figure suggests that up to a CAPE threshold of 5000 J kg$^{-1}$ only minor differences in zonal mean e90 distributions arise.

26. l. 360: The authors mention an 'improvement' of the ECP simulations. What is the chosen reference to improve, how is this quantified?

27. l. 362-365: Parts of this paragraph rather belong in the introduction. Please streamline.

28. l. 373 ff: I find this paragraph rather suitable for Section 3.2 (Statistics of explicitly resolved and parametrized convective updrafts) as it does not relate to tracer distribution but considers convective event frequencies (see comment above to l. 344).

29. l. 383 ff: I find this information quite relevant and would appreciate if it could be placed in the Methods (see general comment 2).

30. l. 390 ff: Could the authors please elaborate on why sensitivity tests are performed without ECP although results suggest substantial under-estimation of convective transport without ECP regardless of the driving dataset (which is also expected given that the applied re-analyses apply convection parameterization).

31. l. 411: Typo: 'It requires only on two input [...]': Remove 'on'.

32. l. 417: Please rephrase 'possible improvement' (see comment above on l. 360).

33. l. 420-440: Given the properties of the underlying wind fields, this is expected. Please see general comments.

34. l. 440-441: Please rephrase.

35. General comment on Conclusions: I would appreciate a thorough and differentiated discussion of the sensitivity experiments. For example, on the relevance of convective thresholds for local vs. global frequencies of convective events, and for the representation of tracer distributions locally versus globally averaged.

---

## Author Comment (AC1)

**Reply to reviewer comments**

Dear Reviewers, dear Editor,

thank you for the time and effort spent on the manuscript and for providing helpful suggestions. We considered all comments and hope that the revised draft properly addresses the open issues. Please find our point-by-point replies below (colored in blue). A revised manuscript with tracked changes has also been prepared.

Best regards,

Lars Hoffmann, on behalf of the co-authors

**Anonymous Referee #1**

This is a well written paper describing the implementation of the extreme convection parameterization into the MPTRAC model and sensitivity simulations describing the impact of using this parameterization on tracer transport. My recommendation is for this paper to be published with minor edits.

Thank you for the encouraging statement.

It is mentioned in Section 2.2 that MPTRAC is used to investigate free troposphere and stratosphere transport processes. It does not mention why it is not used for boundary layer applications until the end of Section 3.7. This should be mentioned in Section 2.2 followed by a discussion of the importance of the depth the tracers are initialized near the surface so they cover the depth of the PBL. I recommend moving Section 3.7 into or soon after Section 2.2.

Following a comment of Reviewer #2, the introductory paragraph of Sect. 3.7 on the sensitivity test regarding the depth of the surface layer was moved forward to the description of the methods in Sect. 2. We kept the remainder of Sect. 3.7 in its current place as we would like to retain our initial ordering in which the information is presented in the manuscript. In Sect. 2.2, we added the following statement to further clarify: "It is important to note that MPTRAC is not targeting boundary layer applications. At present, the model lacks proper representation of more complex mixing and diffusion processes in the boundary layer and it applies pressure as vertical coordinate, which is not terrain-following and therefore less suited for the boundary layer."

Throughout the paper, meteorological model output and reanalysis products are referred to as data. I prefer the word 'data' be reserved for measurements. I suggest re-wording this throughout the paper. For example, I suggest changing the second sentence of the abstract to something like: "Lagrangian transport simulations driven by meteorological fields from global models or reanalysis products, such as the European Centre for Medium-

Range Weather Forecasts' (ECMWF's) ERA5 and ERA-Interim reanalysis, typically lack proper explicit representations of convective up- and downdrafts because of the limited spatiotemporal resolution of the meteorology.

We rephrased the text throughout most of the manuscript as suggested.

Introduction, page 2, line 27: add comma between conditions and such: "...severe weather conditions, such as ..."

Fixed.

Throughout the paper it is mentioned that trajectories were performed. I suggest rephrasing that $10^6$ trajectories were performed to a dispersion simulation with $10^6$ particles. I think of a trajectory as following the mean wind whereas a dispersion simulation includes a turbulent component.

We revised the manuscript to ensure the terms "trajectory calculations" and "dispersion simulations" are properly applied.

Throughout the Results section (especially toward the beginning), be abundantly clear that ECP simulations is a simulation with a CAPE threshold of 0. Same for all of the figure captions.

We agree that the specific values of $CAPE_0$ being used for the ECP simulations need to be made more clear and repeatedly added this information in the text and the figure captions.

Section 3.2, page 11, line 249 – page 12, line 250: Is this shown that it is similar to Konopka et al. (2022)?

We did not conduct a quantitative comparison. We intended to point out that the results are "qualitatively similar" to Konopka et al. (2022). We added the information that our results need to be compared to Fig. 7 of Konopka et al. (2022) to make it more easy for the reader to follow.

Section 3.6: Suggest changing the title of this section: Change "Improvement" to "Sensitivity"

Changed as suggested.

**Anonymous Referee #2**

General comment

This study quantifies convective mass and tracer transport using the Lagrangian MPTRAC model driven by re-analysis data and coupled to the extreme convective parameterization ECP. The ECP randomly redistributes air parcels and tracers between the surface and the equilibrium level if a certain CAPE threshold is exceeded. The authors quantify 'explicit'

versus parametrized convective updrafts and evaluate the choice of the required CAPE threshold to initiate convective transport. Moreover, the sensitivity of artificial tracer distribution to different settings of CAPE, CIN, vertical mixing and surface layer depth is evaluated.

The manuscript is well-structured, comprehensive and well-written. My major concern addresses the stated key research question and conclusion. The key question (l. 62 ff) addresses the question whether ERA5 (compared to ERA-Interim) is sufficient to properly represent convective updrafts and global tracer transport. My concern is that per definition, one would not expect ERA5 (nor ERA-Interim) to adequately represent convective features based on Lagrangian trajectory calculation from the resolved wind fields, as ERA5 uses a convection parameterization and does not represent convective updrafts explicitly. Thus, it is not surprising that ERA5 without ECP does not 'explicitly' resolve a lot of convective features, and requires a parameterization to efficiently vertically mix air parcels and tracers. Thus, the key conclusion (e.g., l. 18 ff and 466 ff) that ERA5 still needs a convection parametrization to better represent convective and tracer transport is expected given the properties of the re-analysis data. Moreover, it is expected that the spatially and temporally coarser ERA-Interim represents less convective features, which has also been already stated by Hoffman et al. 2019 (l. 41 ff and 422 ff). I think the manuscript could profit substantially from focusing on the sensitivity tests and idealised tracer experiments, the effects of implementing CIN, and more directly setting their results into perspective with other convective transport models, instead of comparing ERA5 and ERA-Interim without ECP, as well as ERA5 with and without ECP, which for the reasons mentioned provides expected results. Instead, I would suggest to streamline the manuscript focusing on sensitivity experiments and systematically evaluating their impact on global tracer distribution. Furthermore, the sensitivity tests and the influence of the tuning parameters CAPE and CIN on convective event frequency are shown and discussed. The results are somewhat expected: For example, if the threshold of CAPE to trigger convective mixing is increased, the frequency of convection is reduced in regions that are climatologically characterized by lower CAPE values. Yet, it remains unclear to me which of the tested ECP setting is more realistic beyond the expected impact. Concerning this aspect, the study may benefit from setting their results into perspective with other studies, observations and comparison to more sophisticated convective parameterizations. In summary, I think the manuscript (including abstract, conclusion and results) could benefit from focusing on key aspects of their sensitivity studies and streamlining the sensitivity experiments and novel key results. I have several additional questions and remarks to the authors that are listed below.

Thank you for your detailed feedback and sharing your concerns about the study. We rephrased the stated key research question and conclusions following the suggestions and guidance provided in the major comments. Overall, we think that keeping the simulation results for ECP and non-ECP as well as ERA5 and ERA-Interim is necessary to really provide a clear and complete picture of the implications of applying the ECP in Lagrangian transport simulations. While one might claim it is already known that ERA-Interim has

significantly less explicit convective updrafts than ERA5, it might not be clear how this difference affects individual Lagrangian transport simulations such as those conducted here with and without the ECP for the artificial tracer e90. We found that some authors and recent studies refer to Hoffmann et al. (2019) trying to argue that since the representation of explicitly resolved convective events in ERA5 was improved compared to ERA-Interim, one might not have to care about convection parametrization in Lagrangian transport simulations so much anymore. By systematically comparing ERA5 and ERA-Interim ECP and non-ECP simulation results in this study, we would like to clarify this aspect and advocate for further application of Lagrangian convection parametrizations for tropospheric simulations. We also think it is an interesting finding of our study that despite the fact that there is ten years of development and improvements of the forecast model and data assimilation scheme from ERA-Interim to ERA5, the Lagrangian transport simulation results including the ECP are rather similar between the different data sets. Demonstrating this requires looking at both ERA5 and ERA-Interim. Overall, we agree that streamlining the paper as outlined in the major comments below is very helpful and we tried to mostly follow the suggestions.

Major comments

1 Introduction

Although the introduction reads well, I find the content is rather general and recent studies and developments are not sufficiently explained. For example, l. 54 ff state that "various techniques and parametrizations have been developed to better represent the effects of convection in Lagrangian transport simulations". However, no details are provided which makes it difficult to understand the novelty of this study and to set results into perspective. I would ask the authors to provide a more in-depth overview of current research (which is to some extent referred to later in the manuscript).

We agree that a more in-depth overview on recent work on Lagrangian convection parametrizations will be helpful for the reader to better understand the concept and results of our study. We therefore added three new paragraphs to the introduction of the paper which introduce the different concepts and potential benefits of the Lagrangian convection parametrizations of Brinkop and Jöckel (2019); Konopka et al. (2019); Wohltmann et al. (2019). We also added a rather brief, initial explanation of the main idea of the ECP, which will allow the reader to relate this method to other current schemes.

2 Methods

I would ask the authors to include the first paragraph concerning the sensitivity to the depth of the surface layer into the Methods section (first paragraph in Section 3.7). Moreover, the different settings of the MPTRAC simulations (Sections 3.1, 3.2 as well as 3.3 l. 270-296) could also be placed in a subsection of the Methods. This study uses different settings for the ECP simulations (e.g., CAPE, CIN, vertical mixing, boundary layer depth), which could already be briefly introduced in the Methods part.

Following this comment, we moved most of the initial descriptions and introductory paragraphs on the model settings and test configurations of the different simulations from Sect. 3. to a new Sect. 2.4. This new section allows the reader to more easily compare the different settings and streamlines the results section as requested.

I was wondering why the authors chose different settings for trajectory seeding? Does seeding on a regular global 0.3x0.3 grid introduce biases in convective mass transport as the seeds are not equidistant (for the tracer simulations a latitude weight was applied)? For convective transport the 0.3x0.3 grid was also applied to the coarser ERA-Interim data. In the case of no ECP parameterization (i.e. no random vertical re-distribution), does this result in trajectories that are quasi-identical in ERA-Interim?

For the illustrative example in Sect. 3.1 we added a statement saying that "These parameter settings might be considered representative for a typical application of the ECP." For the analysis of the statistical updrafts, we rephrased "The trajectory seeds were distributed on an $0.3° \times 0.3°$ longitude-latitude grid matching and fully covering the horizontal resolution of ERA5." Here, the idea was to ensure that trajectories were launched in each longitude-latitude grid box of the data, following the approach of Konopka et al. (2022). For the e90 transport simulations, we rephrased: "In the horizontal, the density of the air parcel was weighted with cosine of latitude, to achieve a quasi-homogeneous distribution of the air parcels and the mass." Even though the same sets of trajectory seeds are applied and without the ECP method being considered, ERA5 trajectories will differ from ERA-Interim trajectories.

3 Comparison of ERA-Interim and ERA5

I think the authors could focus on the sensitivity tests with different parameters / threshold for the ECP, instead of comparing ERA5 to ERA-Interim without ECP. It is not surprising that convective updrafts occur less frequently in the coarser ERA-Interim data. Moreover, as also mentioned in l. 41 ff and 422 ff Hoffmann et al. (2019) already showed that "ERA5 explicitly resolves peak updrafts more frequently than ERA-Interim". Moreover, trajectory calculations are hardly comparable due to the different spatiotemporal resolution, in particular, the coarse temporal resolution of ERA-Interim of 6 h. Is it correct that ERA-Interim trajectories are calculated from only two timesteps (as 6-h trajectories are used)? I would thus suggest to focus on the comparison of different ECP settings, and how this affects global tracer distribution.

We tried to clarify the main aim of the study in the introduction section of the paper and rephrased: "In this study, we assess the effects of applying the ECP on Lagrangian transport simulations to properly represent global tracer transport in the free troposphere and stratosphere. Noting that convective transport in the troposphere is generally underestimated in coarse-resolution, global reanalysis horizontal wind and vertical velocity fields driving the Lagrangian transport simulations, the ECP is expected to mitigate these limitations. We conduct our assessment of the ECP using two ECMWF reanalyses, the state-of-the-art ERA5 reanalysis and its predecessor ERA-Interim, in order to evaluate

how the ECP simulations are affected by the different driving meteorological fields. In particular, by systematically comparing ECP and non-ECP simulations with ERA5 and ERA-Interim, we aim to show that while there are large differences in explicitly resolved convective transport between ERA5 and ERA-Interim, both of which significantly underestimate the amount of convective transport in the real atmosphere, the ECP largely mitigates these problems by contributing significantly larger numbers of parameterised convective updrafts to the transport simulations, to a level comparable between ERA5 and ERA-Interim." As we think it would not be clear, whether/how the ECP affects Lagrangian transport simulations using different meteorological input data, we would like to keep the results for both, ERA5 and ERA-Interim, in the paper.

4 Results

In addition to the comments above, I believe that the Results Section could be improved by streamlining and rearranging some paragraphs. Please see also specific comments below.

We streamlined the results section following the advise given in major comment #2 and the specific comments as listed below.

Specific comments and technical corrections

1. l. 24 ff: Please streamline, and include a few relevant references (e.g., for convective mass transport into the stratosphere).

This introductory paragraph is meant to highlight in a broad and general sense the key role of convection in atmospheric dynamics. We streamlined it following the comments on l30 and l32. For the convective mass transport from the boundary layer into the UT/LS we added references to (Dickerson et al., 1987; Fischer et al., 2003; Monks et al., 2009).

2. l. 30: The statement to shallow convection could be removed as it is not relevant in this study.

We removed the statement as suggested.

3. l. 32: Instead of referring to textbooks, the authors could include references about the convection parameterization that is applied in ERA5 and ERA-Interim.

We replaced the references (Smith, 1997; Plant and Yano, 2015) by (Tiedtke, 1989; Kain and Fritsch, 1993; Bechtold et al., 2004, 2008, 2014).

4. l. 48 ff: This paragraph outlines that coarser re-analysis data sets are not capable to explicitly resolve small-scale convective features for various reasons. My question is why the authors use ERA5 and ERA-Interim without ECP as it is expected that convection is underestimated in both data sets? Instead, I would suggest to focus on the ECP settings required to obtain reasonable results. See also general comments.

We rephrased the introduction to try to clarify the aim and objectives of the study and to provide the rationale for assessing both ERA5 and ERA-Interim data. Assessing both, ERA5 and ERA-Interim, allows us to demonstrate that ECP simulation results are rather

similar whereas non-ECP simulations largely differ in the amount of explicitly resolved convective transport. Please see reply to major comment #3.

5. l. 99 f: Please add the number of resulting pressure levels and the approximate vertical spacing in the lower troposphere where trajectories are seeded.

The following information is provided in the new Sect 2.4: "In the baseline simulations, we chose a layer depth of 150 hPa with respect to the surface pressure, corresponding to a layer depth of about 1.1 km at the standard pressure of 1013.25 hPa. It covers nine pressure levels of the ERA-Interim and 19 levels of the ERA5 meteorological fields as prepared for use with the MPTRAC model (see Sect. 2.1)."

6. l. 118: 'particle data': Does this refer to air parcels? Please clarify.

We rephrased this as "air parcel data" to clarify.

7. l. 124 f: How was the timescale of 6 h as typical convective timescale determined? Please clarify and/or add references.

We rephrased this to "...was selected as 6 h to match the order of typical convective timescales..." to indicate that the timescale of 6 h should not be interpreted as a fixed value but might range from just less than an hour to about half a day. We added references to Keil et al. (2014) and Bullock et al. (2015), who discuss the convective adjustment time-scale over which CAPE is being decomposed in convective processes and a reference to Konopka et al. (2022), following which we selected the specific value of 6 h.

8. Figure 2a and l. 175 ff: Does the occurrence frequency show convective events and/or the frequency distributions of CAPE values or is this the same, since convective events are triggered simply by the CAPE threshold. Please clarify.

This is the same. We rephrased "...shows occurrence frequencies of convective events and the frequency distributions of CAPE values exceeding a given threshold..." to clarify.

9. l. 177 ff: Given equation 1, a relation between CAPE and the height of the EL is given. Regarding the still considerable scatter in Fig. 2b, I'm not sure if CAPE only is a reliable quantity to estimate EL (e.g., for a CAPE value of 100 J kg-1 the EL height varies between 3 and 10 km).

We agree that there is still large scatter and added "However, as various additional processes and parameters affect the individual distributions of CAPE and EL, the idea needs to be further investigated in future work." It is no further relevant for our present study as we calculated both, CAPE and EL, directly from the reanalysis temperature and humidity fields and did not exploit the correlation.

10. l. 196: Why did the authors seed trajectories at random vertical positions in the boundary layer and not at various fixed heights?

We added "... to achieve quasi-homogeneous coverage of the layer".

11. l. 200: How was the time interval for ECP of 3 h determined? For subsequent analysis, I believe a threshold of 180 s was used (l. 152). Does this apply to all systematic ECP simulations? How sensitive are simulations to this threshold? I would appreciate if this was discussed in the Methods Section.

For the illustrative example, we mainly selected a lower event frequency (3 h time interval) simply for practical reasons in order to produce a more steady and clear video animation in the supplement. Applying the convection parametrization at each time step of the model (every 180 s) causes the air parcels to be redistributed again and again in the convective columns as long as CAPE is present. Unless the aspect of "upward mixing" is considered (Sect. 2.2), the final outcome of having lower or higher event convective event frequencies is rather similar and may not be need to be discussed in further detail in the manuscript, we think.

12. l. 235: Could the authors please elaborate on the spatial downsampling? Are wind fields averaged, smoothed, etc.?

We added "The methodology of downsampling applied here is described in more detail by Hoffmann et al. (2019)." A more detailed description of the methodology from that paper reads "The process of downsampling or decimation to reduce the sampling rate of a signal typically consists of two steps (e.g., Lyons, 2010). The first step is to apply a low-pass filter to the original data to avoid aliasing of high-frequency features. Here, we applied smoothing with triangular weights in space and time to achieve this effect. The second step is to subsample the smoothed data on the reduced grid. For example, to downsample ERA5 data from hourly to 2-hourly time intervals, we averaged data of $\{t - 1h, t, t + 1h\}$ for a given time $t$ with weighting factors of $\{0.25, 0.5, 0.25\}$ and kept the smoothed data only at a 2-hourly interval. Sensitivity tests showed that this approach including low-pass filtering may significantly reduce aliasing errors and improve simulation results."

13. l. 248-250: Could the authors please elaborate on the similarities and differences between this study and Konopka et al. (2022).

We did not conduct a quantitative comparison but intended to point out that the results are "qualitatively similar" to Konopka et al. (2022). We added the information that our results need to be compared to Fig. 7 of Konopka et al. (2022) to make it more easy for the reader to follow.

14. Figure 5: I would appreciate if the figures could consistently include labels for the applied CAPE threshold (e.g., as panel 5b).

Following a comment by Reviewer #1, we consistently and more frequently added the specific values of $CAPE_0$ throughout the text of the manuscript and in the captions of the Figures.

15. Figure 6: I apologize if this pertains to our printer, but I can hardly distinguish the colors for (i) ERA5 (1000 J kg-1) and ERA5 (spatial DS) as well as (ii) ERA5 (w/o ECP) and ERA-Interim (with ECP).

We modified the colors and the line types of the curves to make it more easy to distinguish between the different cases.

16. l. 270-296: Did the authors consider to move this paragraph to the Methods Section? See general comment 2.

These paragraphs have been moved to the methods section as suggested.

17. l. 256 ff: Please cite the relevant literature here.

We added references to Gerbig et al. (2003); Stein et al. (2015); Konopka et al. (2019, 2022); Loughner et al. (2021).

18. l. 256 ff: Comment related to vertical mixing: To streamline the manuscript, I believe the effects of vertical mixing could be removed. In particular, as no detailed information on potential tuning and the choice of time interval is provided and as it is not applied afterwards.

We agree that the approach of upward mixing needs further evaluation to be better characterized. Although we were not able to continue evaluating it in the present study, we still consider it an interesting open questions regarding the implementation of the ECP method that should be pointed out.

19. l. 265 and 354: Please consistently replace 'equilibrium level' by 'EL'.

Fixed.

20. l. 300: Please rephrase 'vertical layering' (e.g., stratification).

Rephrased as suggested.

21. l. 315 f: Given the property of the underlying wind fields, this general statement is expected.

To be more specific, we added: "This shows that even state-of-the-art reanalyses such as ERA5 with much improved spatiotemporal resolution compared to earlier reanalyses require a convection parametrization in Lagrangian transport models to properly represent transport from the planetary boundary layer into the free troposphere."

22. Figures 7 and 8: I think it could be sufficient to show January and July (e.g., move April and October to Appendix / Supplement) and show differences between Fig. 7 and 8 (similar to Fig. 9 showing differences between ERA5 and ERA-Interim), as all sub-panels look very similar.

We agree that there is not too much additional information from showing the months of April and October, so we removed them and merged Figs. 7 and 8 into one. Figures 9 and 10 were also changed accordingly. This helps to streamline and shorten the results sections as it removes two extra figures.

23. l. 338: 'other months show similar results': They could be added to an Appendix (see

also comment above).

We decided to not include these extra figures in the paper.

24. l. 344 ff: Based on the global distribution of CAPE it is expected that the frequency of convective events decreases with increasing CAPE threshold. Please streamline this paragraph and/or move it to Section 3.2 (Statistics of explicitly resolved and parametrized convective updrafts).

This paragraph and related Fig. 12 are showing convective event occurrence frequencies for July 2017 ERA5 data and were included to better understand the local distributions of the parametrized convective events. We would like to keep this paragraph and Fig. 12 in the results sections.

25. l. 352 f: 'Moderate to strong convective events [...] play a major role in affecting the zonal mean e90 distributions throughout the troposphere': Based on Fig. 11a, I disagree with this statement. The figure suggests that up to a CAPE threshold of 5000 J kg-1 only minor differences in zonal mean e90 distributions arise.

We rephrased the sentence to "Strong convective events with CAPE values of $2000\,\mathrm{J\,kg^{-1}}$ or more play a major role in affecting the zonal mean e90 distributions throughout the troposphere." We initially selected the value of $1000\,\mathrm{J/kg}$ based on deviations of the 90 ppbv contour of e90 from the tropopause in the northern hemisphere extratropics, but rephrasing this towards strong convective effects seems more appropriate.

26. l. 360: The authors mention an 'improvement' of the ECP simulations. What is the chosen reference to improve, how is this quantified?

We revised the wording regarding the improvement of the ECP method throughout the paper and now mostly refer to a "modification" of the method. We moved the description of the proposed modification of the ECP to Sect. 2.3 to streamline the results section. The improvement due to the CIN threshold is summarized in the conclusions: "Considering the modification of the ECP method, we conclude that introducing the threshold $CIN_0$ to hinder the occurrence of parametrized convective updrafts for warm, stable layers, yields local improvements in areas with climatically large CIN (e. g., Northern Africa and Arabian Peninsula)."

27. l. 362-365: Parts of this paragraph rather belong in the introduction. Please streamline.

This paragraph has been moved to the methods section.

28. l. 373 ff: I find this paragraph rather suitable for Section 3.2 (Statistics of explicitly resolved and parametrized convective updrafts) as it does not relate to tracer distribution but considers convective event frequencies (see comment above to l. 344).

We added this paragraph and the corresponding figure in the subsection on the sensitivity test on the parameter $CAPE_0$ as it shows how the choice of $CAPE_0$ affects the distribution of the parametrized convective events.

29. l. 383 ff: I find this information quite relevant and would appreciate if it could be placed in the Methods (see general comment 2).

The paragraph has been shifted to the methods section as suggested.

30. l. 390 ff: Could the authors please elaborate on why sensitivity tests are performed without ECP although results suggest substantial underestimation of convective transport without ECP regardless of the driving data set (which is also expected given that the applied re-analyses apply convection parameterization).

We added: "The sensitivity test for the non-ECP simulations indicates that estimates of convective mass flux from a near-surface layer into the free troposphere will strongly depend on the depth of the layer." This is fundamentally different from the ECP simulations, as discussed in the following paragraph.

31. l. 411: Typo: 'It requires only on two input [...]': Remove 'on'.

Fixed.

32. l. 417: Please rephrase 'possible improvement' (see comment above on l. 360).

Please see reply on comment on l360.

33. l. 420-440: Given the properties of the underlying wind fields, this is expected. Please see general comments.

This paragraph summarizes the finding on the analysis of the explicitly resolved and parametrized convective updrafts. We think it might not necessarily be clear or to be expected how the statistics looks like and how they differ between the two different generations of the ECMWF reanalyses and how the ECP affects them. See responses to related comments.

34. l. 440-441: Please rephrase.

We rephrased this as we addressed the following comment.

35. General comment on Conclusions: I would appreciate a thorough and differentiated discussion of the sensitivity experiments. For example, on the relevance of convective thresholds for local vs. global frequencies of convective events, and for the representation of tracer distributions locally versus globally averaged.

The initial version of the manuscript comprised two paragraphs discussing the results of the sensitivity tests. We revised these paragraphs and split them into three, to separately discuss the impact of $CAPE_0$, $CIN_0$, and the depth of the surface layer. We also extended the text to discuss the implications and effects of the parameter choices on local versus global scale.

**References**

[revised manuscript text omitted]